# Fast Inference for Augmented Large Language Models

**Rana Shahout**
Harvard University

**Cong Liang**
Tsinghua University

**Shiji Xin**
Harvard University

**Qianru Lao**
Harvard University

**Yong Cui**
Tsinghua University

**Minlan Yu**
Harvard University

**Michael Mitzenmacher**
Harvard University

## Abstract

Augmented Large Language Models (LLMs) enhance standalone LLMs by integrating external data sources through API calls. In interactive applications, efficient scheduling is crucial for maintaining low request completion times, directly impacting user engagement. However, these augmentations introduce new scheduling challenges: the size of augmented requests (in tokens) no longer correlates proportionally with execution time, making traditional size-based scheduling algorithms like Shortest Job First less effective. Additionally, requests may require different handling during API calls, which must be incorporated into scheduling.

This paper presents *MARS*, a novel inference framework that optimizes augmented LLM latency by explicitly incorporating system- and application-level considerations into scheduling. *MARS* introduces a predictive, memory-aware scheduling approach that integrates API handling and request prioritization to minimize completion time. We implement *MARS* on top of vLLM and evaluate its performance against baseline LLM inference systems, demonstrating improvements in end-to-end latency by 27%-85% and reductions in TTFT by 4%-96% compared to the existing augmented-LLM system, with even greater gains over vLLM. Our implementation is available online (code, [n. d.]).

## 1 Introduction

Recent progress in large language models (LLMs) has initiated a new wave of interactive AI applications. A prominent example is OpenAI's ChatGPT (OpenAI, 2022), which facilitates conversational interactions across various tasks. One way to extend LLM capabilities is to augment them with external tools (Mialon et al., 2023), resulting in what we refer to as API-augmented requests. These augmentations include arithmetic calculation (Hao et al., 2024), ChatGPT plugins (OpenAI., 2023), image generation (Betker et al., 2023), and virtual environments (Shridhar et al., 2020). Consequently, AI development is increasingly moving towards compound AI systems (Zaharia et al., 2024) that integrate multiple interacting components, such as model calls, retrievers, and external tools, rather than relying solely on monolithic models.

API-augmented requests present several challenges, particularly regarding memory consumption during the LLM decoding phase, which is memory-bound. Each request has associated key and value matrices that grow in size during the request. LLMs cache these matrices in a key-value (KV) cache throughout the sequence generation process to enhance efficiency, eliminating the need to recompute them at every iteration. This caching significantly reduces computation time but requires substantial memory. High memory consumption during decoding can translate to higher latency

39th Conference on Neural Information Processing Systems (NeurIPS 2025).

and lower throughput, as it limits the system's ability to process multiple requests concurrently. (Appendix A includes the background for LLM execution and API interactions.)

With API augmentation, memory demands can increase further based on how the system manages requests during API calls. There are three primary memory handling strategies: *Preserve*, which retains the KV cache in memory while awaiting the API response; *Discard and Recompute*, which discards the KV cache and recomputes it once the API returns; and *Swap*, which offloads the KV cache to CPU memory and reloads it after the API call. Each strategy has its drawbacks. Preserve leads to excessive memory usage, Discard and Recompute incurs additional computational overhead, and Swap introduces latency due to data transfers.

Existing LLM inference systems are primarily designed for standalone LLMs and struggle to maintain low request completion times for augmented LLMs. A key issue that can lead to delays is head-of-line (HoL) blocking, where long-running requests, including those awaiting API responses, prevent shorter ones from being processed efficiently. Scheduling strategies can reduce HoL by prioritizing requests. Traditional size-based scheduling prioritizes jobs[1] with shorter execution times, which works well for standalone LLM requests where execution time correlates with output size (Shahout et al., 2024; Fu et al., 2024). However, in API-augmented requests, output size no longer reliably indicates total request time, as a request with a short output might involve a lengthy API call, while a longer output might require minimal API interaction.

To address this challenge, we propose integrating the scheduling and memory handling of requests rather than treating LLM execution and API calls as separate processes. We introduce *MARS* (*Memory- and API- Rooted Scheduler*), a novel inference framework designed to optimize augmented LLM latency. *MARS* utilizes two steps: (1) assigning a handling strategy to API-augmented requests **prior to scheduling**, based on predictions of output size and API call duration, and (2) scheduling requests by ranking them according to their **predicted total memory**, which estimates the memory footprint across a request's lifecycle, factoring in both request size and API interactions.

INFERCEPT (Abhyankar et al., 2024), a closely related system, dynamically classifies API-augmented requests into handling strategies (preserve, discard, or recompute) **only when a request reaches its API call**. However, it relies on first-come, first-served (FCFS) scheduling, which may cause HoL blocking and hinder latency guarantees. In contrast, *MARS* integrates handling strategy assignment with scheduling by leveraging predictive information. This allows us to schedule requests based on our newly proposed predicted (remaining) total memory. While it is theoretically possible to collectively optimize handling and scheduling decisions, such an approach is impractical in an online setting. Consequently, we employ a greedy algorithm that first minimizes memory usage for each individual request and then schedules requests based on their total memory requirements.

Our approach begins with a prediction model that estimates pre-API output sizes from input prompts and predicts API durations by API type, enabling us to assign each request's handling strategy before scheduling. We then introduce a scheduling policy that ranks requests according to their predicted total memory, thereby minimizing request completion time. We integrate optimizations into our scheduling policy to mitigate starvation and reduce scheduling overhead. Finally, we build our system on top of vLLM (Kwon et al., 2023), a state-of-the-art LLM inference system. We evaluate *MARS* on three datasets, comparing *MARS* against baseline systems. Our results show that *MARS* consistently outperforms INFERCEPT and vLLM across various datasets and request rates, achieving improvements in end-to-end latency ranging from 27% to 85% and reductions in TTFT from 4% to 96%, with even greater improvements over vLLM. We also analyze the components of *MARS* and the effect of prediction accuracy on its performance. To ensure scalability, we designed *MARS* to support multi-GPU setups, which was validated through testing with Llama 70B.

## 2 The Challenge of Scheduling API-augmented Requests

Swapping KV cache to host memory at API calls may appear straightforward but incurs hidden costs. Even with full PCIe bandwidth, transfers bottleneck foreground tasks and block new requests. Large and fragmented contexts add overhead from multiple kernel launches. Despite pipelined swaps Abhyankar et al. (2024), bandwidth remains a limit. INFERCEPT shows swapping still

---

[1]The terms job and request are used interchangeably in this paper.

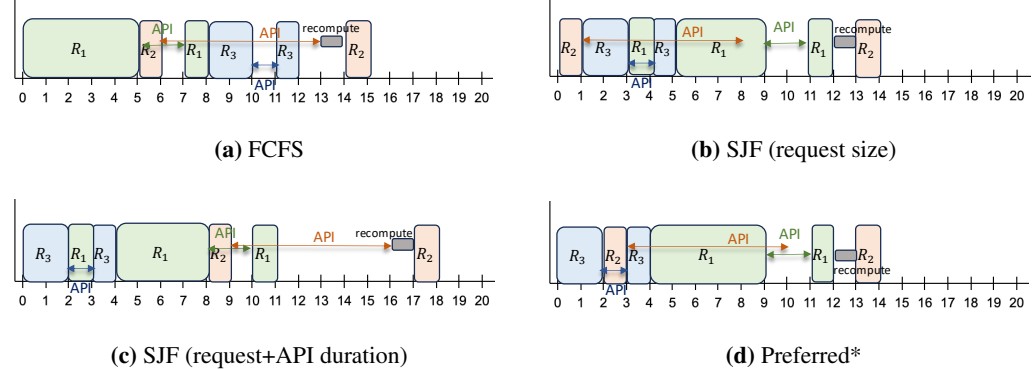

**Figure 1:** Comparison of scheduling policies for API-augmented requests with a memory budget of 6 units. Average completion times: (a) FCFS: 11.66 units, (b) SJF: 10.33 units, (c) SJF (size + API): 11 units, (d) Optimized: 10 units.

wastes 26% of GPU resources and 25% of runtime in mixed workloads. Thus, no single handling strategy fits all requests.

Our goal is to minimize average and typical request times by scheduling requests across different handling strategies. Size-based scheduling methods, such as Shortest Job First (SJF), can reduce request completion time by utilizing known or predicted request sizes (where, in this context, size specifically refers to the execution time needed to finish the job). Traditional scheduling methods, however, encounter challenges with API-augmented requests. Indeed, it is not clear what the appropriate notion of "size" should be with API calls for size-based scheduling: should the API delay be included or not?

**Example.** Consider three requests R1, R2, and R3 that all arrive at time 0. Each request includes one API call, triggered at different times during decode generation. Their output sizes are 6, 2, and 3 tokens, respectively, with API durations (in token generation units) of 2, 7, and 1 units, respectively. The strategies to handle requests during the API for each request are Preserve for R1, Discard for R2, and Swap for R3.

In this example, we assume that only one request can run at a time and the memory budget is limited to 6 units.

The strategy for handling each request during its API call was determined dynamically at runtime using the INFERCEPT equations (1,2 and 3 in (Abhyankar et al., 2024)). For simplicity, we assume complete information is known about the total size and API duration of each request. A request is scheduled only if there is enough memory available.

Figure 1 shows different scheduling policies for this example. Although all requests arrive at the same time, the FCFS scheduling policy used by INFERCEPT determines their order based on request ID, processing them as $R_1, R_2, R_3$. With a memory budget of 6 units, the scheduler processes the pre-API part of $R_2$ during $R_1$'s API call, as $R_2$ will be discarded after one unit, freeing memory to continue with $R_1$. In contrast, $R_3$'s pre-API part cannot run during $R_1$'s API call because it will not release memory before the API response completes, preventing $R_1$ from resuming. This scheduling yields an average request completion time of 11.66 units (Figure 1(a)). The SJF policy schedules requests based only on size, processing them from shortest to longest: $R_2, R_3, R_1$. At time unit 8, the API of $R_2$ completes, leaving a post-API part of size 2 (including recomputation). However, the running request, $R_1$, also has two units remaining, so $R_2$ must wait. At time unit 9, $R_1$ enters its API call, consuming five units of memory, leaving only 1 unit available, which is insufficient to start the post-API part of $R_2$. As a result, $R_2$ must wait until $R_1$ finishes. This policy results in an average request completion time of 10.33 units.

These approaches fall short of optimal scheduling because they ignore the interaction between scheduling and request handling during API calls. A naive strategy, referred to as *SJF by total size* (Figure 1(c)), orders requests by output size plus API duration. Again, in this example, the pre-

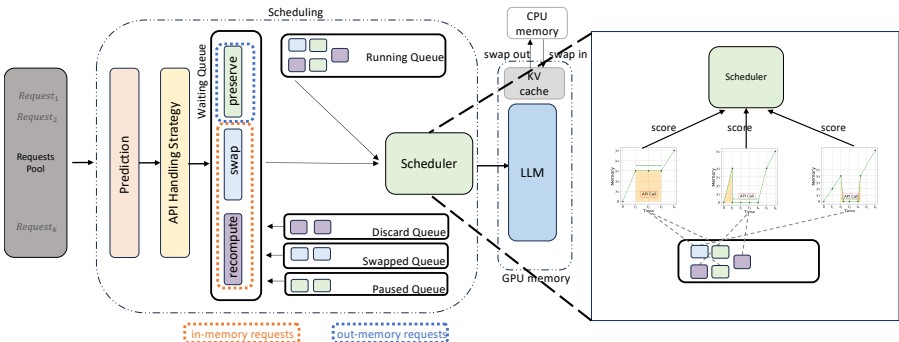

**Figure 2:** *MARS* architecture. *MARS* minimizes completion time for API-augmented requests through two steps: determining request handling strategies (preserve, discard, or swap) to minimize memory waste by predicting pre-API output size and API properties before scheduling, and implementing a scheduling policy based on the handling method and request output size.

API part of $R_2$ can run during $R_1$'s API call. This policy achieves an average request completion time of 11 units, worse than SJF. A more effective scheduling policy (Figure 1(d)) integrates total size and the API handling strategy. Our intuition is that, under memory constraints, $R_3$, the least memory-intensive request, should run first to release memory quickly. It should be followed by $R_2$, with $R_1$ (the most memory-consuming request due to its Preserve handling) scheduled last. This approach yields an average request completion time of 10 units, outperforming previous methods. Notably, the post-API part of $R_2$ becomes ready at time unit 10, but due to memory constraints, it waits until $R_1$ finishes.

The insight from this example informs our proposal to incorporate API handling strategies into the scheduler, ranking requests based on their total memory consumption.

## 3  *MARS* Design

**Problem Formulation.** Let $\mathcal{R}$ be a set of API-augmented requests. Each request $r \in \mathcal{R}$ is modeled as an ordered list of tuples corresponding to its API calls. Given a limited KV memory budget, the goal is to minimize the average end-to-end latency of requests, defined as the time from submission to completion, while satisfying GPU memory capacity constraints by assigning an API handling strategy per request and determining the optimal processing order.

**System Overview.** *MARS* employs two main steps to reduce LLM inference response time: (1) determining the handling strategy for requests during API calls **prior to scheduling** to minimize memory waste by predicting the pre-API output size and API duration, and (2) implementing a scheduling policy that considers both request size and the chosen handling strategy, ranking requests based on remaining memory consumption.

Figure 2 illustrates the *MARS* architecture. Users submit requests to the request pool. *MARS* predicts pre-API output size and estimates API properties (duration and response size, Table 1 in Appendix B) based on input prompts. Using these predictions, *MARS* estimates total memory consumption, considering this in API handling decisions and scheduling policy ranking. *MARS* determines how to handle requests during API calls to minimize memory waste. Based on this handling method and request output size, *MARS* implements a scheduling policy tailored for API-augmented requests. The pseudocode of the *MARS* scheduler is provided in Algorithm 1 in Appendix B.

**Step 1: Handling Strategy Assignment.** Our goal is to choose the API handling strategy that is predicted to minimize memory waste before a request is processed, enabling the scheduler to rank each request based on a chosen handling strategy. We explain here how to predict the best handling strategy for requests with API calls under a single-API call assumption; extensions to multiple API calls are discussed later in this section. We first predict the output size and API duration. Our approach generalizes beyond the dataset by using a predictor to estimate pre-API output size based on the prompt. Output size prediction has been studied in the context of LLMs (Jin et al., 2023; Stojkovic et al., 2024; Cheng et al., 2024; Shahout et al., 2024). For API duration, we leverage

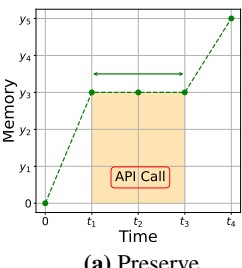 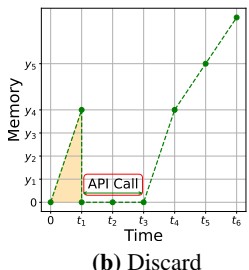 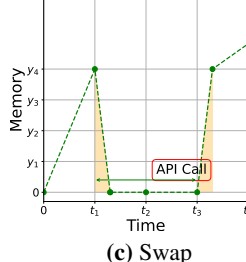

| (a) Preserve | (b) Discard | (c) Swap |

**Figure 3:** Memory consumption for a request over time, from arrival to completion, with one API call using handling strategies: (a) Preserve, (b) Discard and Recompute, and (c) Swap. The highlighted area indicates the memory waste for a single request.

the fact that APIs belong to fixed classes (e.g., Math, Image), each with consistent functionality and a similar duration. By extracting the API type from the prompt, we estimate the API response size using the average size from the training set for that API class. This method relies on the standardized outputs and consistent behaviors of APIs within each class[2]. The handling strategy for an API call is determined by predicting memory waste, using pre-API and API duration estimates. Unlike INFERCEPT, which selects handling strategies only when a request reaches an API call, *MARS predicts the handling strategy in advance,* enabling the integration of strategy assignments with scheduling.

Figure 3 shows the memory consumption of a request over time, from its arrival to completion. We define total memory usage as the integral of memory consumption throughout the request's execution until completion. The shaded regions in Figures 3(a),3(b),3(c) represent wasted memory, which is defined as the integral of memory consumption over the time period during which it is not utilized for actual inference. Memory waste varies depending on the chosen strategy. In the Preserve strategy (Figure 3(a)), memory remains allocated throughout the API call, causing idle memory waste during the wait time. In the Discard strategy (Figure 3(b)), memory is released during the API call, reducing waste but requiring recomputation afterward. In the Swap strategy (Figure 3(c)), memory is temporarily swapped out and reloaded after the API call, resulting in a spike in memory usage when swapping back in. We combine this waste with our estimation of the context size for batched requests according to the setup. Using pre-API and API duration predictions, we estimate the wasted memory for each handling strategy based on the behavior of each strategy, as shown in Figure 3. *MARS* then selects the strategy that minimizes this total waste.

Instantaneous memory measurements fail to reflect how long memory resources are occupied, which is particularly important in the decoding phase of LLMs. It is not just the amount of memory a request consumes at a particular moment, but also how long that memory remains in use. A strategy that uses more memory for a shorter period can be more efficient than one that uses less memory but occupies it longer.

**Multi-API.** To generalize to requests with multiple APIs, we divide each request into segments, each ending with a single API call. After completing an API call, the request re-enters the system as a new request for the next API call. Each segment is classified based on the current API's characteristics. For example, a job with initial processing followed by two API calls is divided into segments, each consisting of a processing phase and an API call, with the final segment representing the last processing phase. We estimate the returned token size for each segment based on the specific API. While this approach does not account for cumulative memory usage, predicting the total number of API calls and their resource usage is challenging, and is left for future work. This segmentation aligns with INFERCEPT, which processes multi-API requests incrementally as they reach each API.

**Effect of Mispredictions.** Mispredictions are to be expected. Small mispredictions in API duration or output size will typically have a small effect; indeed, they may not change the overall ranking of jobs. Mispredicting a short API or output as long may have a large effect on that particular job,

---

[2]This approach could be further extended by incorporating predictions for API response size, which we leave for future work.

but does not typically harm other jobs in the system (Mitzenmacher, 2021). A long-running API call incorrectly predicted as short may lead the system to select a memory-wasteful strategy, such as Preserve. This unnecessary memory consumption may limit the system's ability to process additional requests. A request with a long output misclassified as short may cause head-of-line blocking and delay other requests. Importantly, better predictions would only further improve system performance by reducing these misclassifications. At the same time, using simple averages from training provides a lightweight, low-overhead predictor that we show already yields practical benefits. Improving prediction accuracy remains a promising future work.

**Step 2: Scheduling Policy through Ranking.** Once a handling strategy is assigned to each request, the second step of *MARS* focuses on scheduling the requests. Minimizing request completion time requires a unified scheduling method that considers both the total size of requests and their specific handling strategies during API calls. For example, it may order two requests with the same total size based on the handling strategy during the API call. Or it may prioritize a request with a longer total size but a more memory-friendly handling strategy over a shorter request with a handling strategy that may more negatively impact system performance. Intuitively, requests should be ranked based on their memory consumption. We emphasize that without API calls, ranking based on memory consumption aligns with ranking based on execution time (or request size), as memory consumption has a linear relationship with request size. With API calls, this relationship breaks, and a request's handling strategy is selected to minimize memory consumption during the API. Consider Figure 3, which shows the total memory; we consider the area (integral) as a rank function of a request and select the function based on the predicted handling strategy during the API call. Referencing Figure 1, among the three requests, $R_3$ consumes the least memory and should be prioritized, followed by $R_2$. $R_1$ consumes the most memory due to its size, API duration, and the preserve handling strategy, so it should be scheduled last.

**Starvation Prevention.** Scheduling policies can cause certain requests to experience long wait times, leading to high tail latency, a form of starvation that degrades system performance and user experience. This issue arises when longer or resource-intensive requests are continually deferred in favor of shorter ones, exacerbating tail latency. Our memory-focused scheduling policy alone does not detect and mitigate starvation, which can result in extended wait times and reduced fairness. To solve this, we have implemented a starvation prevention mechanism to improve the scheduler's tail latency using a per-request counter. The counter increments when a request remains in the waiting queue for a new iteration. Upon reaching a predefined threshold, *MARS* tags the request as starving and prioritizes it by placing it at the head of the scheduled requests for the current iteration. The relative order of prioritized and non-prioritized requests is maintained according to *MARS*'s ranking decisions. Prioritization continues until request completion, avoiding memory waste from preempted (half-finished) requests. If the request has not been prioritized and encounters API calls or is scheduled, the counter resets to 0. Parameter experiments led us to set the predefined threshold at 100 (testing with the datasets in Section 4). Additionally, while SRPT scheduling is often considered as unfair, (Bansal and Harchol-Balter, 2001) shows that SRPT can achieve better fairness compared to FCFS under many practical conditions.

**Handling mixed workloads of API- and non-API requests.** *MARS* naturally handles mixed workloads by ranking non-API requests based on their estimated memory consumption (i.e., prompt and output size), which is equivalent to execution-time ordering due to their near-linear relationship, consistent with existing LLM scheduling methods Fu et al. (2024); Shahout et al. (2024). Appendix C.2 presents evaluations on mixed workloads of API and non-API requests, showing that *MARS* consistently reduces mean response time compared to both INFERCEPT and vLLM.

**Comparison with INFERCEPT.** While MARS builds on INFERCEPT's concept of measuring memory waste over time, our system is significantly different in at least three aspects: (1) *Use of Predictions:* MARS predicts output sizes and API durations to assign handling strategies before scheduling, efficiently integrating them into the scheduling process. In contrast, INFERCEPT dynamically selects strategies during API calls without proactive scheduling. (2) *Scheduling:* MARS prioritizes requests based on predicted **total** memory requirements, combining scheduling with handling strategy, while INFERCEPT schedules the request by FCFS. (3) *System Optimization:* MARS introduces a starvation prevention mechanism to balance efficiency and fairness.

***MARS* Limitation.** *MARS* currently approximates API output size using average response length. However, actual API responses can be highly variable. In future work, *MARS* could incorporate

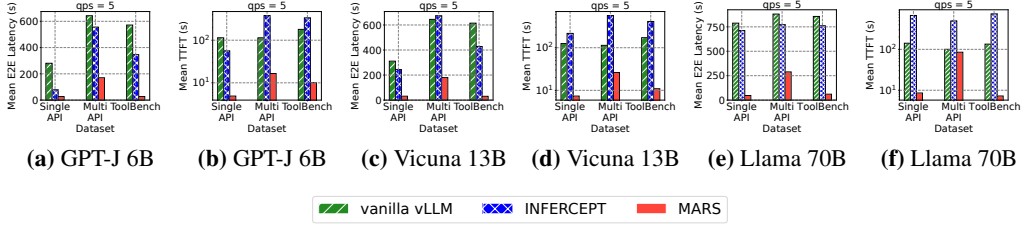

**(a)** GPT-J 6B  **(b)** GPT-J 6B  **(c)** Vicuna 13B  **(d)** Vicuna 13B  **(e)** Llama 70B  **(f)** Llama 70B

vanilla vLLM   INFERCEPT   MARS

**Figure 4:** End-to-end performance (mean and P99 of latency and TTFT) of single-API, multi-API, ToolBench datasets with request arrival rate fixed to five when serving GPT-J 6B and Vicuna 13B.

more accurate response-size predictors. An additional possibility, if API responses are highly variable, is to first do a 1-bit prediction to determine if the output size is short or long. The survey in Mitzenmacher and Shahout (2025) details this approach.

## 4 Evaluation

Our primary evaluation metrics are end-to-end latency (the time from when a request is submitted to the system until its completion) and time-to-first-token (TTFT) (both mean and P99) across three datasets and four model sizes, comparing *MARS*'s performance against INFERCEPT (Abhyankar et al., 2024) and vanilla vLLM (Kwon et al., 2023). In addition, we provide a breakdown of *MARS*'s components to evaluate the contribution of each. Our experiments also evaluate the impact of prediction noise (simulated mispredictions), measure prediction accuracy, overhead, and test different starvation thresholds. *MARS* is implemented on top of vLLM, using the OPT-125M model (Zhang et al., 2022) for predictions. Implementation details are provided in Appendix C.1.

**LLM models.** We use four models with different sizes: the TinyLlama-v1.1 model with 1.1B (results in appendix C.3), the 6B-parameter GPT-J model (GPT-J 6B), the 13B-parameter Vicuna model (Vicuna 13B) and Meta-Llama-3-70B (Llama 70B).

**Testbed.** We used a machine with dual AMD EPYC 7313 CPUs (16 cores each, 64 threads total), 503 GB RAM, and two NVIDIA A100 GPUs (80 GB each) connected via NVLink. For GPT-J 6B and Vicuna 13B, GPU memory usage was capped at 40 GB to match INFERCEPT's setup. Llama 70B was served using vLLM's default tensor parallelism (set to 2) across the two GPUs.

**Datasets.** We evaluate our system using three datasets. The first two, based on INFERCEPT. The single-API dataset is a subset containing only a single API, while the multi-API dataset is the full INFERCEPT dataset. The third dataset, ToolBench (Qin et al., 2023), is an instruction-tuning dataset for tool-use tasks, featuring over 16,000 real-world APIs across 49 categories. We use it to predict output size, API duration, and response size.

**End-to-end Performance.** Figure 4 illustrates how *MARS* consistently achieves lower latency and TTFT across all datasets at a request rate of 5, highlighting its advantage over vLLM and IN-FERCEPT. At this rate, using different model sizes, *MARS* demonstrates improvements across the datasets. Figure 5 shows how varying the request arrival rate affects the mean and P99 of end-to-end latency and TTFT across the three datasets using GPT-J 6B and Vicuna 13B, while Llama 70B results appear in Appendix C. *MARS* consistently outperforms vLLM and INFERCEPT in mean TTFT and end-to-end latency across all tested datasets for both GPT-J 6B and Vicuna 13B. At lower request rates (e.g., 3) on the single-API dataset, *MARS* reduces mean TTFT by around 4–5% compared to INFERCEPT and 18–23% compared to vLLM, while showing mean end-to-end latency improvements of up to 1% over INFERCEPT and 15% over vLLM. As the arrival rate increases (e.g.,4 or 5), these gains become more pronounced: *MARS* achieves TTFT reductions of over 90% and latency improvements of up to 80–90% compared to both baselines. On the multi-API dataset, it similarly reduces TTFT by more than 89% and end-to-end latency by up to 78%. For the ToolBench dataset, *MARS* yields TTFT improvements of 75–99% and latency gains of over 60% compared to INFERCEPT and vLLM at a request rate of 3.

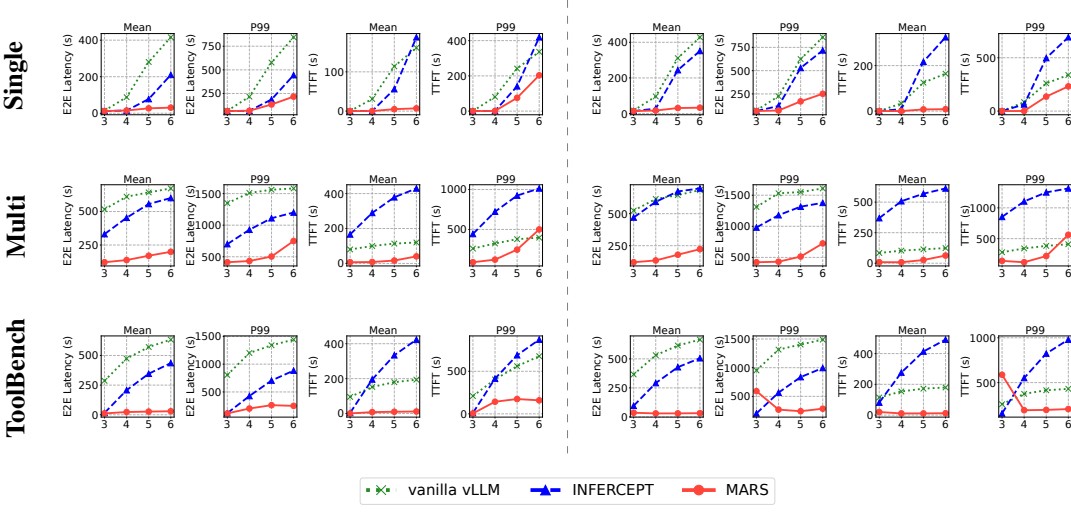

**Figure 5:** End-to-end performance (mean and P99 of latency and TTFT) as a function of request arrival rate when serving GPT-J 6B and Vicuna 13B (results using Llama 70B appear in Appendix C) using different datasets (single-API, multi-API, ToolBench).

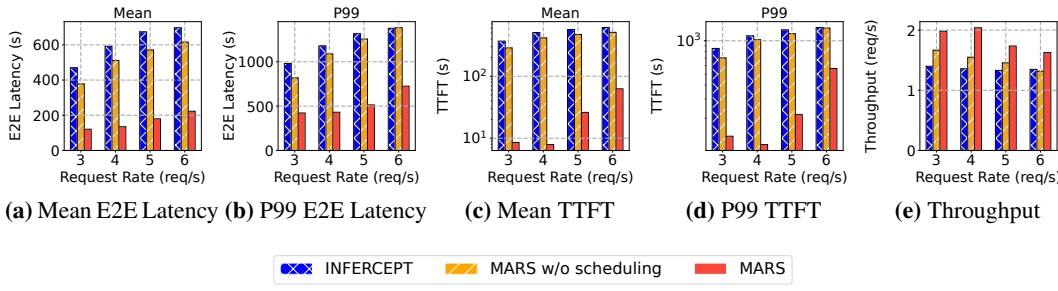

**(a)** Mean E2E Latency **(b)** P99 E2E Latency **(c)** Mean TTFT **(d)** P99 TTFT **(e)** Throughput

**Figure 6:** Breakdown of *MARS* components, using Multi-API dataset with Vicuna 13B.

**Starvation thresholds.** Figure 16 in Appendix C compares the throughput and tail latency of *MARS* under various starvation prevention thresholds.

**Breakdown of *MARS* Components.** To further understand the benefits of *MARS*, we incrementally added its components to vLLM and compared the results with INFERCEPT. We used the Multi-API dataset because it has the highest latency among the datasets, Figure 6 shows throughput, end-to-end latency and TTFT. First, we added the predicted API handling component to vLLM while keeping the scheduling policy as FCFS (referred to as *MARS* w/o scheduling). With this addition, the performance was close to INFERCEPT but slightly worse. The key difference between INFERCEPT and *MARS* w/o scheduling is that *MARS* uses predicted information to pre-determine API handling, while INFERCEPT makes dynamic decisions at the API call. Integrating our scheduling policy brought improvements across all metrics, but API handling predictions remain a crucial prerequisite for implementing the scheduler.

**Effect of Mispredictions.** Using the INFERCEPT dataset, we inject Gaussian errors into API duration and output size predictions: error $\sim \mathcal{N}(0, p \times m)$, where $p$ is the error parameter and $m$ is the measured value. Predictions are calculated as predicted_value = measured_value + error. By varying $p$, we assess how prediction inaccuracies affect *MARS*'s performance. Figure 7(a) shows the impact of prediction errors on end-to-end latency and throughput. As $p$ increases (e.g., 5%,

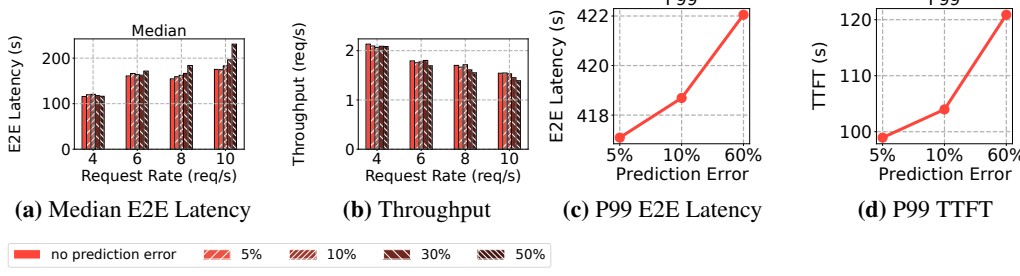

**Figure 7:** Error injection, Multi-API dataset with GPT-J 6B.

10%, 30%, 50%), median latency rises, especially at higher request rates (8–10 req/s), indicating longer waiting times due to inaccuracies. Throughput (Figure 7(b) also decreases at higher error rates, particularly under heavy loads. In Figures 7(c),7(d), we increase error injection further under $req/sec = 3$. The figures show that while higher prediction error raises p99 latency and TTFT, the overall impact remains modest.

**Prediction Accuracy and Overhead.** Appears in Table 2 in Appendix C.

**Memory Occupancy.** Appendix C.7 presents GPU and host memory–occupancy profiles over time.

## 5 Related Works

Several studies focus on improving inference throughput through optimized scheduling strategies. Orca (Yu et al., 2022) introduces iteration-level scheduling, where a new batch is created at the end of each model forward pass. vLLM (Kwon et al., 2023) introduces paged attention, treating the KV cache as virtual memory mapped to non-contiguous physical GPU memory. Another line of work addresses the imbalance between the prefill and decoding stages. Sarathi (Agrawal et al., 2024) employs chunked prefill, which divides prompt tokens into smaller chunks merged with decoding requests to form a batch for each iteration. Splitwise (Patel et al., 2024) separates the prefill and decoding stages across different machines to match their distinct computational demands. These techniques are complementary and can be integrated with *MARS*.

Different approaches have been explored to design effective scheduling policies for LLMs. Fast-Serve (Wu et al., 2023) builds upon Orca by scheduling each output token individually using a Multi-Level Feedback Queue. However, this approach leads to frequent preemptions, increasing the cost of managing the KV cache memory and offloading to the CPU. AQUA Vijaya Kumar et al. (2025) addresses GPU memory contention during LLM inference by proposing network-accelerated memory offloading and a preemptive scheduling framework that reduces paging overheads and enables responsive, high-throughput inference. Prediction-based scheduling methods have been introduced to address these challenges. Trail (Shahout et al., 2024) obtained output size predictions directly from the target LLM by feeding the embedding of its internal structures into a lightweight classifier. LTR (Fu et al., 2024) learns to rank requests based on their output size. Importantly, however, these previous works focus on requests without API augmentations.

## 6 Conclusion

We have introduced *MARS* (*Memory- and API- Rooted Scheduler*), an LLM inference framework designed explicitly for API-augmented requests. *MARS* optimizes request completion time through a unified scheduling strategy that ranks requests based on their total memory consumption, integrated over time. Our approach enables *MARS* to handle varying output sizes and API interactions, with a starvation prevention mechanism to improve tail latency. Experimental results demonstrate that *MARS* improves end-to-end latency by 27%-85% and reduces TTFT by 4%-96% compared to INFERCEPT, with even greater gains over vLLM.

**Impact Statement.** We believe that this work serves as a starting point for API-augmented requests. Generally, we suggest that scheduling with API calls appears to open the door to many interesting algorithmic problems. We are not aware of API calls of the form considered here being studied in the (theoretical) scheduling algorithms literature. More consideration of algorithmic bounds for problems may yield more additional practical strategies for API-augmented requests.

**Acknowledgments.** Rana Shahout was supported in part NSF grants DMS-2023528. Michael Mitzenmacher was supported in part by NSF grants CCF-2101140, CNS-2107078, and DMS-2023528.

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

# Appendix

## Table of Contents

## A  Background

### A.1  Transformer-Based Generative Models

At each step, a Transformer model generates the most probable next token based on the sequence of previously generated tokens. A model generating a sequence of length $n$ needs to perform $n$ iterations, with each token passing through several layers of self-attention and feed-forward networks.

During the $i$-th iteration, the model operates on all prior tokens $(t_0, t_1, \ldots, t_{i-1})$ using self-attention mechanisms. The resulting output can be represented as:

$$h_{\text{out}} = \text{softmax}\left( \frac{q_i \cdot K^\top}{\sqrt{d_h}} \right) \cdot V$$

Here, $q_i$ is the query vector for the current token $t_i$, while $K$ and $V$ are matrices containing the key and value vectors for all preceding tokens, where $K, V \in \mathbb{R}^{i \times d_h}$.

### A.1.1  Key-Value (KV) Cache

To reduce computational overhead, LLMs cache the key and value matrices (KV cache) during sequence generation. This approach avoids recomputing these matrices at each step, improving efficiency but leading to high memory usage, which scales with the sequence length, number of layers, and hidden dimensions. As more tokens are generated, memory demands grow, particularly for long sequences. For instance, the GPT-3 175B model requires around 2.3 GB of memory to store key-value pairs for a sequence length of 512 tokens. This high memory consumption poses challenges for efficient preemptive scheduling, especially when working with limited GPU memory.

## A.2 Augmented LLMs

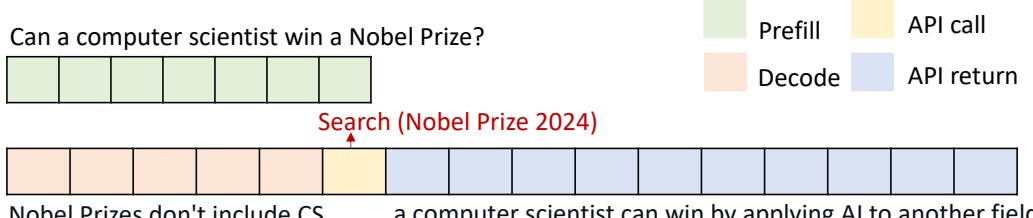

**Figure 8:** Illustration of an augmented-LLM request. The API fetches detailed information about the 2024 Nobel Prize.

Augmented Language Models (Mialon et al., 2023; Wang et al., 2024) refer to language models that enhance their capabilities by incorporating external tools, retrieval mechanisms, or reasoning strategies to overcome the limitations of traditional LLMs. Unlike pure LLMs, which rely solely on pre-trained parameters to generate responses, augmented LLMs can query external data sources to expand their capabilities. Figure 8 shows an example of an augmented LLM request. These augmentations, which we refer to as *API* (Application Programming Interfaces), fall into three main categories as described in (Mialon et al., 2023): incorporating non-LLM tools during decoding (such as calculators (Wolfram, [n. d.]), information retrieval systems (Baeza-Yates et al., 1999)), iterative self-calling of an LLM (like chatbots maintaining conversation history), and complex compositions involving multiple LLMs, models, and tools (exemplified by frameworks like LangChain (Chase, [n. d.]), DSpy (Khattab et al., 2024), Gorilla (Patil et al., 2023), SGLang (Zheng et al., 2023), and AgentGraph (Chen et al., 2019)).

LLM API time varies significantly based on augmentation types, with a clear distinction between short-running and long-running augmentations. Despite this variation, today's systems still rely on FCFS scheduling. This suggests that API handling strategies should be tailored to specific augmentation types rather than using a one-size-fits-all approach.

With API augmentation, memory demands increase further, depending on how the system handles requests during API calls. Figure 9 shows the impact of including API calls: using a subset of INFERCEPT (Abhyankar et al., 2024), we compare two variations of the dataset—one with API calls and one without in terms of KV cache usage (%) over time when all API calls are handled using Preserve and in terms of completed requests number over time using Preserve.

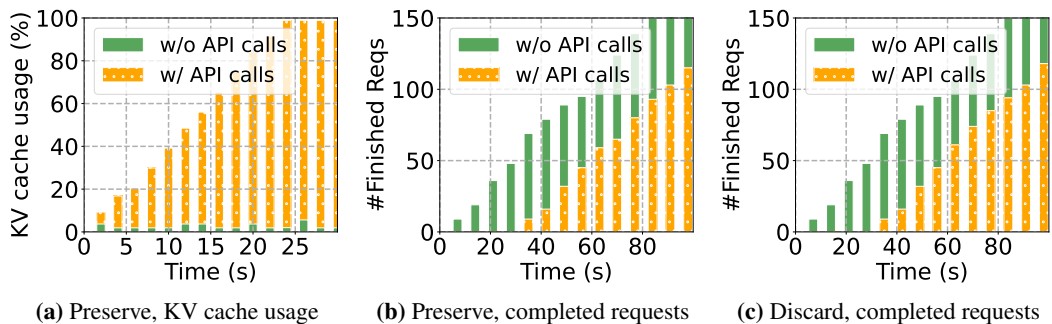

(a) Preserve, KV cache usage    (b) Preserve, completed requests    (c) Discard, completed requests

**Figure 9:** Impact of including API calls: (a) KV cache usage (%) over time when all API calls are handled using Preserve. (b) Number of completed requests over time using Preserve. (c) Number of completed requests over time using Discard.

## A.3 Handling Requests During API

Optimizing memory management during API calls involves selecting a strategy that minimizes GPU memory waste. In an augmented LLM inference system, this choice depends on two factors: the

duration of the API call and the length of the pre-API output. For brief API calls, the Preserve strategy may be advantageous in avoiding recomputation overhead. For longer API calls, either Discard or Swap is preferable. If the pre-API portion of the request is computationally light (short), Discard is beneficial. Otherwise, Swap may be more efficient despite the potential delays it introduces.

INFERCEPT addresses this optimization challenge by developing the following equations (equations 1-3 in (Abhyankar et al., 2024)) that model the memory wastage associated with each handling strategy.

$$\text{WastePreserve}_i = T_{\text{INT}} \times C_i \times M \tag{1}$$

$$\text{WasteDiscard}_i = T_{\text{fwd}}(C_i) \times C_i \times M + T_{\text{fwd}}(C_i) \times C_{\text{other}} \times M \tag{2}$$

$$\text{WasteSwap}_i = 2 \times T_{\text{swap}}(C_i) \times C_{\text{batch}} \times M \tag{3}$$

Here $T_{\text{INT}}^j$ is the duration of the API call for request $i$, and $C_i$, $C_{\text{other}}$, and $C_{\text{batch}}$ represent the context size (in tokens) of request $i$ before the API call, the context size of other requests in the batch with request $i$, and the total context size of all requests in the batch, respectively. $M$ denotes the memory consumed per token for the KV cache. $T_{\text{fwd}}(C_i)$ and $T_{\text{swap}}(C_i)$ represent the time required for model forwarding with context $C_i$ and the time to swap context $C_i$, respectively. INFERCEPT dynamically selects a strategy that minimizes memory waste. However, its scheduling policy remains FCFS.

# B  *MARS* Scheduler

The pseudocode of the *MARS* scheduler is provided in Algorithm 1.

**Algorithm 1** *MARS* Scheduler

---

1: **Input:** Request pool $P$, predictor model $Predictor$, waiting queue $WaitingQueue$, running batch $runningBatch$, starvation threshold $StarvationT$
2: **while** True **do**
3:    **for all** $r \in P$ **do**
4:       $predictions_r \leftarrow Predictor(r.\text{prompt})$
5:       $r.\text{handling} \leftarrow HandlingStrategy(predictions_r)$
6:       $WaitingQueue.\text{put}(r)$
7:    **end for**
8:    **for all** $r \in PQueue \cup DQueue \cup SQueue$ **do**
9:       **if** $r.\text{APIcallFinished}()$ **then**
10:          $WaitingQueue.\text{put}(r)$
11:       **end if**
12:    **end for**
13:    **for all** $r \in WaitingQueue$ **do**
14:       $r.\text{score} \leftarrow HandlingRanking(r)$
15:    **end for**
16:    $WaitingQueue \leftarrow \text{Sort}(WaitingQueue)$ by $r.\text{score}$
17:    $runningBatch \leftarrow \emptyset$
18:    **for all** $r \in WaitingQueue$ **do**
19:       **if** $runningBatch$ not full **then**
20:          $runningBatch \leftarrow runningBatch + r$
21:          $r.\text{StarvationCnt} \leftarrow 0$
22:       **else**
23:          $r.\text{StarvationCnt} \leftarrow r.\text{StarvationCnt} + 1$
24:       **end if**
25:    **end for**
26:    **for all** $r \in WaitingQueue$ **do**
27:       **if** $r.\text{StarvationCnt} \geq StarvationT$ **then**
28:          Place $r$ at $WaitingQueue$ head
29:          $r.\text{StarvationCnt} \leftarrow 0$
30:       **end if**
31:    **end for**
32:    Remove finished requests from $WaitingQueue$
33:    Execute $runningBatch$
34:    **for all** $r \in runningBatch$ **do**
35:       **if** $r.\text{encounterAPIcall}()$ **then**
36:          **if** $r.\text{handling} == Preserve$ **then**
37:             $PQueue.\text{put}(r)$
38:          **else if** $r.\text{handling} == Discard$ **then**
39:             $DQueue.\text{put}(r)$
40:          **else if** $r.\text{handling} == Swap$ **then**
41:             $SQueue.\text{put}(r)$
42:          **end if**
43:       **end if**
44:    **end for**
45: **end while**

---

Table 1 shows API call properties (duration and number) based on their type.

| Dataset | Type | Duration (sec) | Num |
|---------|------|----------------|-----|
| INFERCEPT | Math | (9e-5, 6e-5) | (3.75, 1.3) |
| | QA | (0.69, 0.17) | (2.52, 1.73) |
| | VE | (0.09, 0.014) | (28.18, 15.2) |
| | Chatbot | (28.6, 15.6) | (4.45, 1.96) |
| | Image | (20.03, 7.8) | (6.91, 3.93) |
| | TTS | (17.24, 7.6) | (6.91, 3.93) |
| ToolBench | - | (1.72, 3.33) | (2.45, 1.81) |

**Table 1:** API durations and number for two different datasets: INFERCEPT (Abhyankar et al., 2024) and ToolBench (Qin et al., 2023). First part of this table is taken from INFERCEPT (Abhyankar et al., 2024) (Table 1).

# C   Evaluation

## C.1   Implementation Details

Our implementation supports multi-GPU setups, as evidenced by our evaluation with the Llama 70B model using vLLM's default tensor parallelism across two GPUs. To implement the prediction mechanism, we use the OPT-125M language model (Zhang et al., 2022), a transformer-based model developed by Meta. With 125 million parameters and support for a context length of 2048 tokens, OPT-125M can effectively handle datasets with long contexts, such as the multi-API dataset. Although smaller than many larger language models, OPT-125M delivers strong language generation capabilities. Our approach utilizes the embeddings generated by OPT-125M during the initial processing of input prompts. After tokenizing the input and processing it through the model's layers, we extract the final token's embedding, which is then fed into a linear classifier. This classifier assigns the input to one of 50 bins, each representing a range of 10 tokens, and is trained using cross-entropy loss.

The model estimates the completion length for each prompt based on learned representations from the ToolBench dataset (Qin et al., 2023), which involves complex conversations with API interactions. We train the model using an 80-20 split for training and validation, classifying output lengths into bins. We apply this model specifically to the ToolBench dataset because the other dataset already includes detailed output length information, making prediction unnecessary in that case. *MARS* is evaluated using the test portion of the ToolBench data to ensure accuracy.

## C.2   Mixed workloads of API- and non-API requests.

Figure 10 shows the mean end-to-end latency and TTFT as functions of request arrival when serving Vicuna 13B using a mixed workload of API- and non-API requests. The workload is derived from the InferCept Multi-API dataset, where 50% of the requests are set as non-API (i.e., the API call is disabled but the request is kept). The evaluation demonstrates that *MARS* consistently outperforms INFERCEPT and vLLM on a mixed workload of requests with and without API calls.

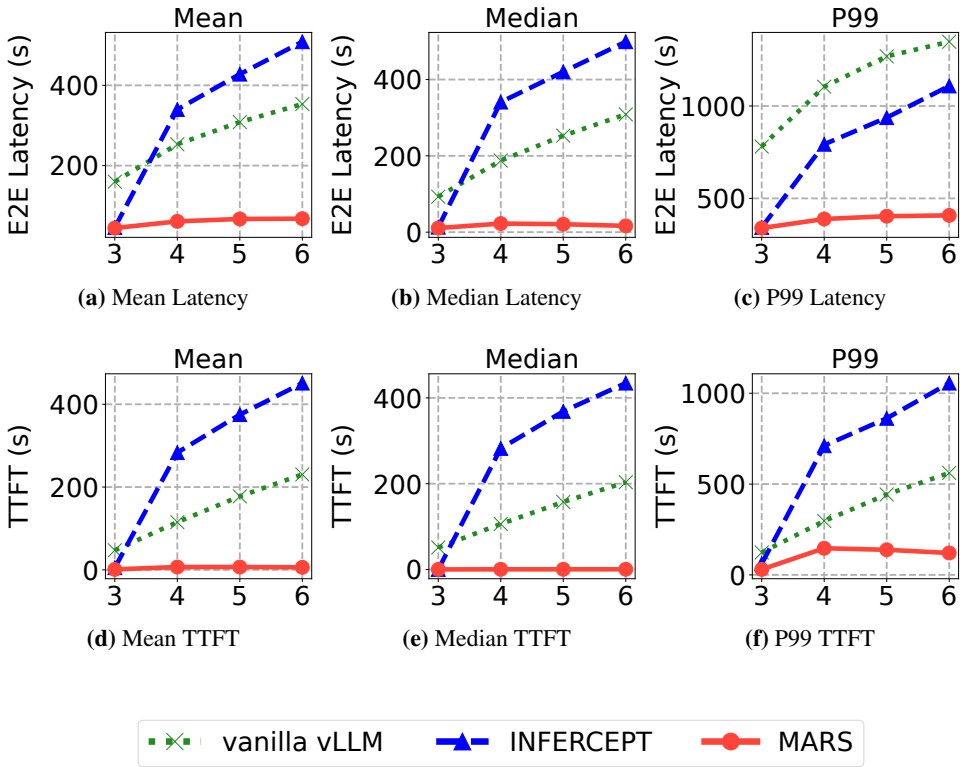

**Figure 10:** End-to-end performance as a function of request arrival rate when serving Vicuna 13B using a mixed workload of API- and non -API requests.

## C.3   TinyLlama v1.1 (1.1B) Results

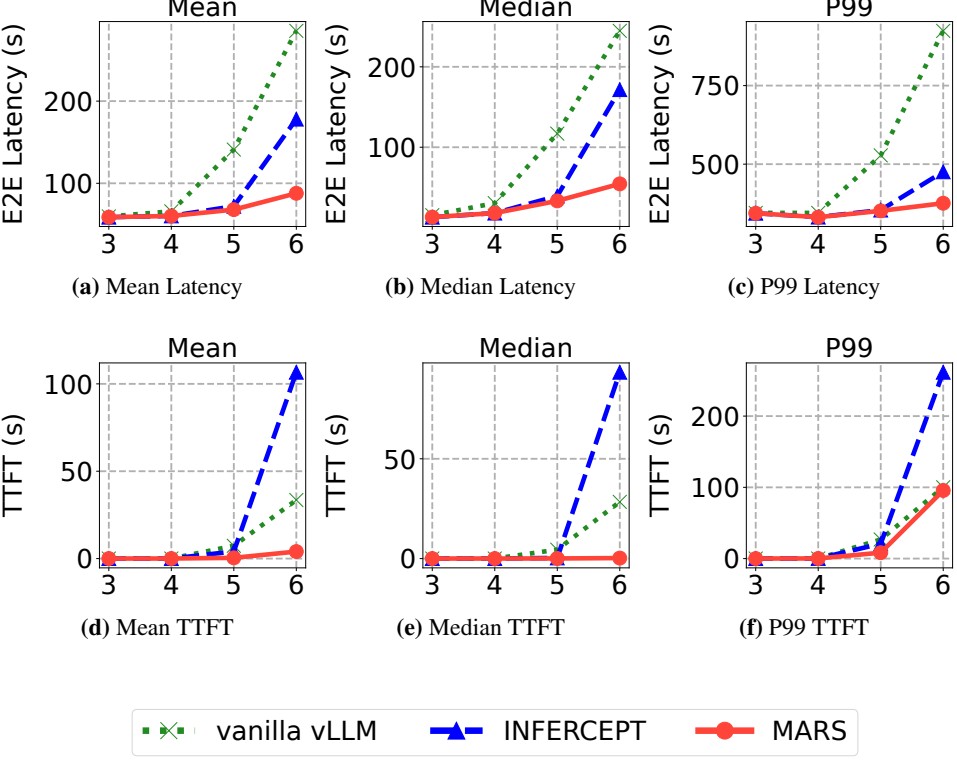

**Figure 11:** End-to-end performance as a function of request arrival rate when serving TinyLlama 1.1B using different INFERCEPT datasets.

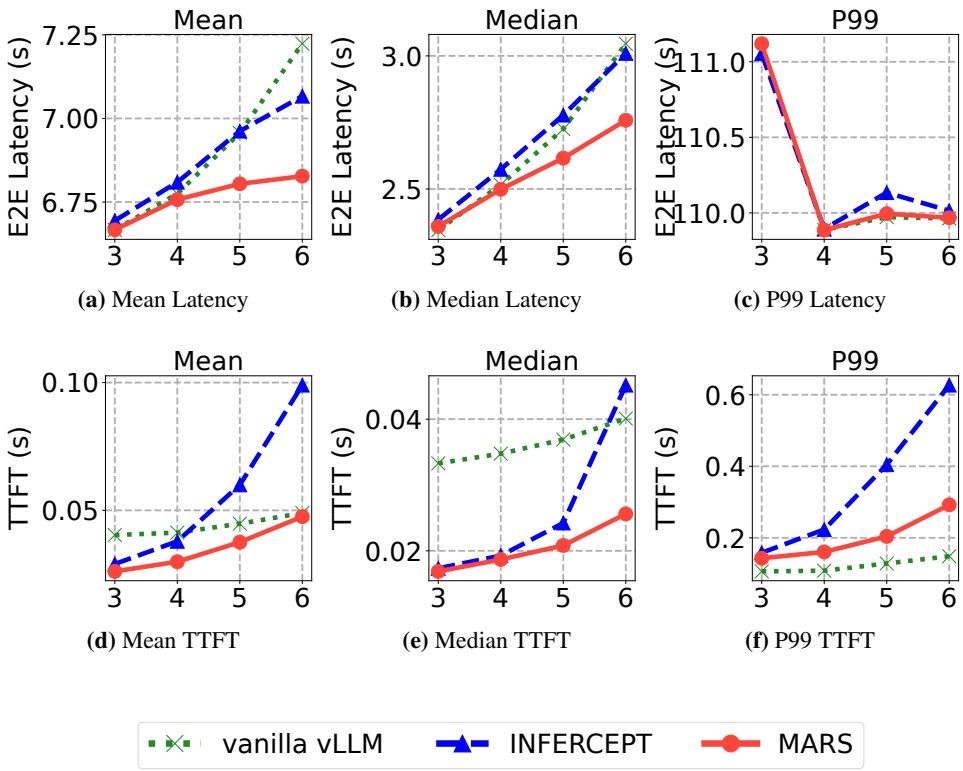

**Figure 12:** End-to-end performance as a function of request arrival rate when serving TinyLlama 1.1B using different Toolbench dataset.

Figure 11 and 12 compare *MARS*, INFERCEPT, and vLLM using the small TinyLlama v1.1 model with 1.1 billion parameters. Specifically, they report the mean, median, and P99 end-to-end latency and TTFT as functions of request arrival rate, using the INFERCEPT and ToolBench datasets, respectively. The results show that *MARS* consistently improves both end-to-end latency and TTFT over vLLM and INFERCEPT.

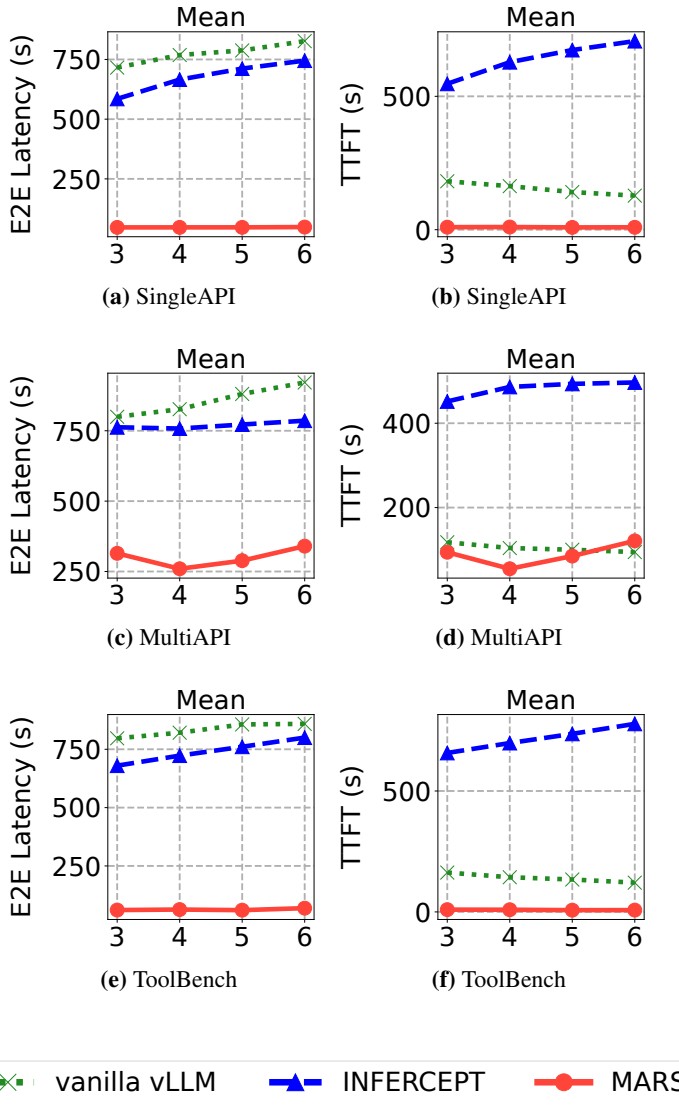

**Figure 13:** End-to-end performance as a function of request arrival rate when serving Llama 70B using different datasets (single-API, multi-API, ToolBench).

Figure 13 illustrates the mean end-to-end latency and TTFT as functions of request arrival rate across three datasets: Single API, MultiApi, and ToolBench, when serving Llama 70B. The results demonstrate that *MARS* achieves improvements in both end-to-end latency and TTFT compared to vLLM and INFERCEPT.

## C.5 Starvation thresholds

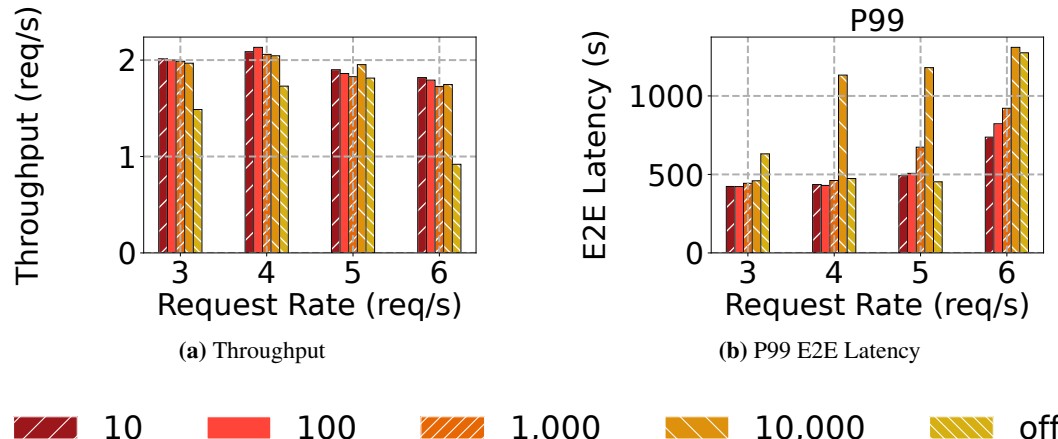

**(a)** Throughput

**(b)** P99 E2E Latency

10     100     1,000     10,000     off

**Figure 16:** Starvation threshold, Multi-API dataset with GPT-J 6B.

Figure 16 compares the throughput and tail latency of *MARS* under various starvation prevention thresholds. We observe that introducing a starvation prevention threshold reduces tail latency and enhances throughput, with a threshold of 100 providing a good balance between both metrics.

## C.6 Prediction Accuracy and Overhead.

| Bin num | 0 | 1 | 2 | 3 | 4 | 5 | 6 | 7 | 8 | 9 | 10 |
|---------|-----|------|-------|-------|-------|-------|-------|-------|-------|-------|-----|
| Acc-5 | 0.0 | 0.75 | 0.834 | 0.813 | 0.713 | 0.769 | 0.568 | 0.5 | 0.321 | 0.333 | 0.3 |
| Acc-15 | 0.0 | 0.75 | 0.921 | 0.933 | 0.867 | 0.851 | 0.691 | 0.545 | 0.321 | 0.462 | 0.3 |

**Table 2:** Bin accuracy for top 10 bins.

We evaluated the precision of our response length predictions using the ToolBench dataset by measuring the absolute difference between the predicted and actual word lengths (which is part of the dataset). Table 2 shows the results.

We used two accuracy metrics, Acc-5 and Acc-15 that represent the percentage of predictions that differ from the actual length by no more than 5 words and 15 words, respectively. The results show 68.5% accuracy for Acc-5 and 78.3% accuracy for Acc-15, with a Mean Absolute Error (MAE) of 3.06. When focusing on the first 20 bins (responses up to 200 words), the MAE improves to 1.366, indicating higher accuracy for shorter responses. We used an NVIDIA A100 GPU for inference, achieving an average prediction time of 13.7 ms per input on the ToolBench dataset. Table 2 shows Acc-5 and Acc-15 per bin for the first ten bins.

## C.7 Memory Occupancy

Figure 17 GPU and host memory occupancy by replaying a mixed API workload on the Vicuna 6B model using the multi-API INFERCEPT dataset. We record the amount of KV cache resident on the GPU and any swapped–out context held in host memory.

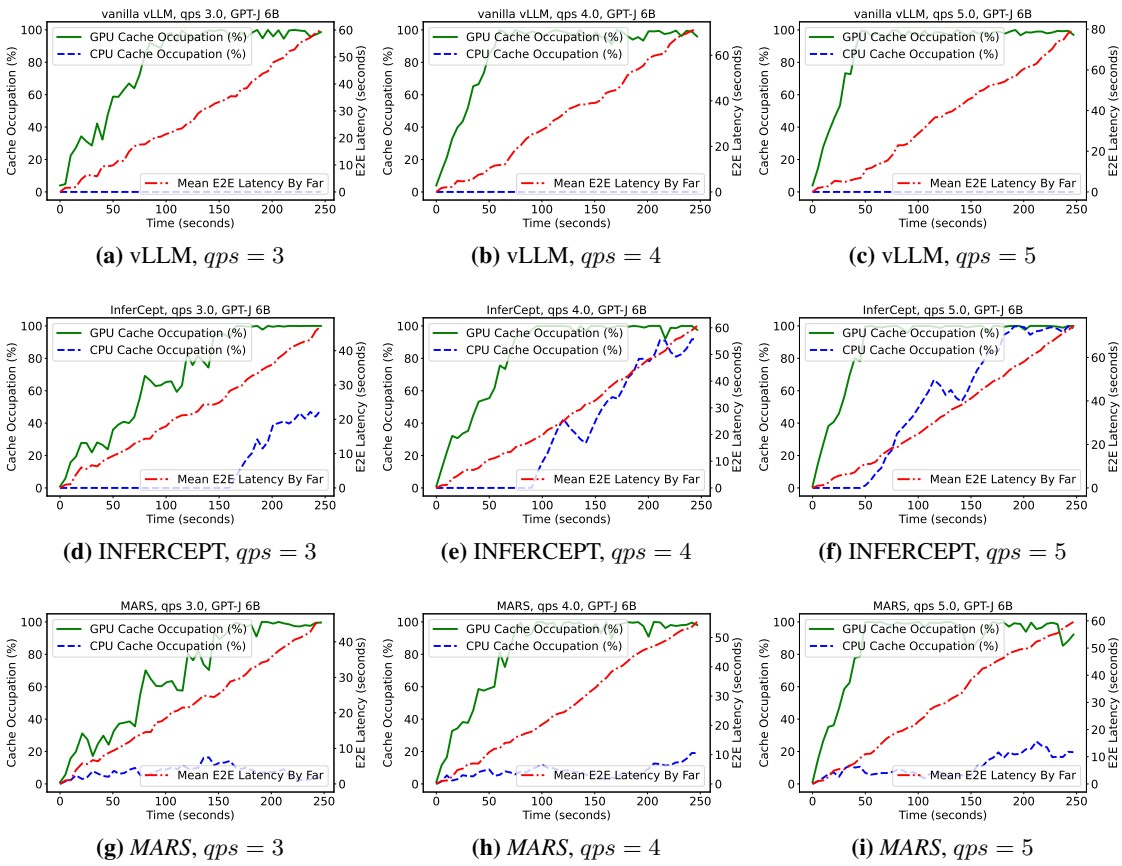

**Figure 17:** GPU/CPU cache occupation and mean end-to-end latency over time on the Vicuna 6B model using the multi-API INFERCEPT dataset.

