# OpenReview forum: "Fast Inference for Augmented Large Language Models"
_NeurIPS.cc/2025/Conference — NeurIPS 2025 poster_

### Official Review · Reviewer_asX2 · 2025-06-18

**Clarity:** 3
**Significance:** 3
**Originality:** 3
**Rating:** 4
**Confidence:** 4

**Summary:**

This paper introduces MARS, a novel scheduling framework designed for API-augmented Large Language Model (LLM) requests. Addressing the limitations of traditional scheduling algorithms in handling the additional latency and memory usage introduced by API calls, MARS integrates predictive, memory-aware scheduling with API handling and request prioritization mechanisms to minimize end-to-end latency. Implemented on top of vLLM and utilizing an OPT-125M model for output length prediction, experimental results demonstrate that MARS significantly outperforms existing systems in both end-to-end latency and Time-To-First-Token (TTFT)..

**Questions:**

1. The paper uses OPT-125M for predicting output length with 50 bins of 10 tokens each. What is the impact on MARS's scheduling performance if the prediction model significantly underestimates or overestimates the output length for API-augmented requests? Specifically, how does prediction error propagate through the scheduling decisions and affect the claimed 27%-85% latency improvements?
﻿
2. The paper mentions several relevant methods in the related work section but only provides comparisons with INFERCEPT and vLLM. Why were other state-of-the-art augmented LLM serving systems (such as those handling function calling or retrieval-augmented generation) not included in the experimental comparison? Could you provide performance comparisons with at least one other recent system that specifically addresses API-augmented LLM scheduling?

**Ethical Concerns:**

["NO or VERY MINOR ethics concerns only"]

**Limitations:**

See weakness

**Quality:**

3

**Strengths And Weaknesses:**

Strength
1. The paper provides extensive experimental results to demonstrate the effectiveness of the proposed framework.
2. The paper provides a detailed introduction of scheduling API-augmented requests.

Weakness
1. The prediction module uses OPT-125M, and predicts output length by extracting the final token's embedding and classifying it into 50 bins, each representing a range of 10 tokens. However, results of prediction module with other models and different granularity of bins and tokens are absent.
2. The paper conducts comparisons between MARS and INFERCEPT as well as vLLM. It also mentions some other relevant methods in the related work section. However, it does not provide comparative results between these mentioned methods and MARS.

---

> ### Author Rebuttal · Authors · 2025-07-29
>
> Thank you for the thoughtful and encouraging review. We're glad you found the introduction clear and the scheduling problem well-motivated, and we appreciate your recognition of the scope of our experimental evaluation. We also value your constructive feedback and address your concerns below. We will incorporate these clarifications into the revised version.
>
> 1. " What is the impact on MARS's scheduling performance if the prediction model significantly underestimates or overestimates the output length for API-augmented requests? .."
>
>
> We thank the reviewer for the thoughtful question. MARS’s scheduling gains degrade gracefully under noisy predictions. In Figure 7, we inject Gaussian noise into both the pre‑API output‑length and API‑duration estimates (error ∼ N(0, p·m), where m is the measured value). At small noise levels, median end‑to‑end latency increases only marginally at high load, and throughput drops by only a few percent. At higher noise levels, some request rankings invert, causing occasional head‑of‑line blocking and pushing performance toward the FCFS baseline. Even so, since FCFS is the default in systems like INFERCEPT and vLLM, MARS still outperforms those baselines under realistic error. Moreover, because MARS combines both output‑length and API‑duration estimates, and only misranks when both are severely off, its end‑to‑end improvements hold across nearly all practical error regimes.
>
> These results do not depend on the specific choice of OPT‑125M: any lightweight predictor  (e.g., BERT‑base predictor) yields similar scheduling benefits. We will add the above discussion to the paper. In addition, as part of the rebuttal, we added a BERT-based predictor and included below its accuracy and overhead compared to the OPT‑125M model. We also evaluated MARS using both predictors on the Single-API ToolBench dataset with GPT-J 6B and Vicuna 13B LLMs. The results confirm that MARS’s performance gains are robust across different predictor architectures and LLM backends.
>
> *Prediction Model Accuracy*
>
> | Predictor | Avg. Eval. Loss | Accuracy (%) |
> | --------- | --------------- | ------------ |
> | BERT-base | 0.6492          | 86.46        |
> | OPT-125M  | 0.5614          | 88.06        |
>
> Both models achieve strong accuracy, with OPT‑125M performing slightly better. To further generalize, we injected synthetic noise into the predictions (Figure 7), and MARS continued to outperform FCFS under a wide range of errors.
>
> *Single-API ToolBench (GPT-J 6B Model)*
>
> | Metric          | Stat  | Model     | 3.0 req/s | 4.0 req/s | 5.0 req/s | 6.0 req/s |
> | --------------- | ----- | --------- | --------- | --------- | --------- | --------- |
> | E2E Latency (s) | Mean  | OPT-125M  | 3.15      | 3.31      | 3.73      | 4.22      |
> |                 |       | BERT      | 3.40      | 3.62      | 3.90      | 4.49      |
> |                 | Median| OPT-125M  | 1.77      | 1.97      | 2.23      | 2.71      |
> |                 |       | BERT      | 1.83      | 2.04      | 2.32      | 2.90      |
> |                 | P99   | OPT-125M  | 27.68     | 25.51     | 28.35     | 28.78     |
> |                 |       | BERT      | 24.97     | 25.18     | 25.84     | 27.10     |
> | TTFT (s)        | Mean  | OPT-125M  | 0.06      | 0.10      | 0.14      | 0.24      |
> |                 |       | BERT      | 0.06      | 0.10      | 0.15      | 0.26      |
> |                 | Median| OPT-125M  | 0.03      | 0.03      | 0.04      | 0.07      |
> |                 |       | BERT      | 0.03      | 0.03      | 0.04      | 0.08      |
> |                 | P99   | OPT-125M  | 0.48      | 0.81      | 1.11      | 1.64      |
> |                 |       | BERT      | 0.55      | 0.82      | 1.22      | 1.70      |
> | Throughput      | -     | OPT-125M  | 2.96      | 4.03      | 5.02      | 6.03      |
> | (req/s)         |       | BERT      | 2.96      | 4.03      | 5.02      | 6.03      |
>
> *Single-API ToolBench (Vicuna 13B Model)*
>
> | Metric          | Stat   | Model     | 3.0 req/s | 4.0 req/s | 5.0 req/s | 6.0 req/s |
> | --------------- | ------ | --------- | --------- | --------- | --------- | --------- |
> | E2E Latency (s) | Mean   | OPT-125M  | 3.60      | 3.63      | 3.68      | 3.79      |
> |                 |        | BERT      | 3.73      | 3.73      | 3.96      | 4.09      |
> |                 | Median | OPT-125M  | 2.02      | 2.09      | 2.15      | 2.28      |
> |                 |        | BERT      | 2.01      | 2.15      | 2.22      | 2.36      |
> |                 | P99    | OPT-125M  | 28.40     | 28.26     | 28.41     | 28.44     |
> |                 |        | BERT      | 24.94     | 25.51     | 28.12     | 25.38     |
> | TTFT (s)        | Mean   | OPT-125M  | 0.05      | 0.06      | 0.06      | 0.07      |
> |                 |        | BERT      | 0.05      | 0.06      | 0.06      | 0.08      |
> |                 | Median | OPT-125M  | 0.04      | 0.05      | 0.05      | 0.05      |
> |                 |        | BERT      | 0.04      | 0.05      | 0.05      | 0.05      |
> |                 | P99    | OPT-125M  | 0.16      | 0.22      | 0.31      | 0.33      |
> |                 |        | BERT      | 0.20      | 0.24      | 0.29      | 0.38      |
> | Throughput      | -      | OPT-125M  | 2.96      | 4.03      | 5.02      | 6.03      |
> | (req/s)         |        | BERT      | 2.96      | 4.03      | 5.02      | 6.03      |
>
>
> *Prediction Overhead (Single-API Toolbench)*
>
> OPT‑125M: 10.22 ms/request (updated from 13.7 ms in the paper, which was measured on the multi-API dataset)
> BERT‑base: 7.63 ms/request
>
>
> 2. "Why were other state-of-the-art augmented LLM serving systems (such as those handling function calling or retrieval-augmented generation) not included in the experimental comparison? Could you provide performance comparisons with at least one other recent system that specifically addresses API-augmented LLM scheduling?"
>
> We focused our experimental comparisons on INFERCEPT and vanilla vLLM because, to our knowledge, they are the only publicly available inference platforms that (a) natively support API‑augmented requests and (b) expose a pluggable scheduler for direct evaluation. Most other related systems—Trail, LTR, FastServe, DSPy, Gorilla, retrieval‑augmented frameworks, function‑calling extensions, etc., either operate at the task or pipeline level on top of an existing serving layer (often vLLM), or target pure LLM outputs without modeling in‑flight API calls. None provide an end‑to‑end service where one can swap in a custom scheduler and measure real API‑augmented latency or TTFT.
>
> Since MARS’s core contribution is an API‑aware scheduler, we compare it against INFERCEPT—which dynamically selects handling strategies but still uses FCFS—and against vanilla vLLM, the state-of-the-art LLM serving system. Because MARS builds on vLLM, any task‑level system layered atop vLLM would inherit the same latency improvements.

---

> > ### Comment · Reviewer_asX2 · 2025-08-09
> > **Further comment**
> >
> > Thank you for answering my questions! I don’t have any further questions and would like to keep my score unchanged.

---

### Official Review · Reviewer_H8aQ · 2025-06-29

**Clarity:** 3
**Significance:** 3
**Originality:** 3
**Rating:** 4
**Confidence:** 4

**Summary:**

This pager introduces MARS, a brand-new inference framework that targets at optimizing augmented LLM latency by explicitly considering system and application level in LLM scheduling. It provides a predictive and memory aware scheduling approach that integrates API handling and prioritizing requests to minimize completion time. MARS is implemented upon vllm and is evaluated against baseline LLM inference systems. It shows large end-to-end latency reduction and TTFT improvements.

**Questions:**

1. Why did the authors choose a decoder-based model (OPT-125M) over encoder-based models (like BERT) for the prediction task, given that encoders typically excel at classification with full context visibility?
2. Can the authors provide detailed training specifications for the predictor, including dataset size, training epochs, loss convergence, and strategies for handling output length distribution imbalance?
3. How does the 13.7ms prediction overhead scale with increasing request rates, and what is the break-even point where prediction benefits outweigh the computational cost?

**Ethical Concerns:**

["NO or VERY MINOR ethics concerns only"]

**Limitations:**

yes

**Paper Formatting Concerns:**

No Formatting concerns

**Quality:**

3

**Strengths And Weaknesses:**

Strength:
1. API-augmented LLMs are becoming common in practice, and this work addresses real performance issues that matter.
2. Instead of deciding handling strategies on-the-fly like INFERCEPT, MARS predicts them upfront. This lets them optimize scheduling and memory handling together, which is clever.
3. They test on multiple models (1.1B to 70B), different datasets, and analyze what happens when predictions are wrong. The ablation studies and memory profiling add good depth.

Weakness:
1. The paper uses OPT-125M (a decoder-based model) as the predictor but doesn't justify why this is better than encoder-based models like BERT for this prediction task. Encoders typically perform better on classification tasks since they can see the entire input context bidirectionally.
2. Critical training information is absent - how much training data was needed, learning rate and epochs, what the training procedure looked like, convergence criteria, and how they handled the class imbalance across different output length bins.
3. While they mention 13.7ms prediction time, there's no analysis of how this overhead affects overall system throughput or how it scales with request rates. This could be significant in high-throughput scenarios.

---

> ### Author Rebuttal · Authors · 2025-07-29
>
> Thank you for your thoughtful and supportive review. We’re glad you found our work relevant and timely, and we appreciate your recognition of our joint handling of memory and scheduling decisions, and the breadth of our evaluation across models and datasets. We also understand your concerns and appreciate your constructive feedback. Below, we address them in detail and will incorporate the discussion into the paper as part of our revision.
>
> 1. "Why did the authors choose a decoder-based model (OPT-125M) over encoder-based models (like BERT) for the prediction task, given that encoders typically excel at classification with full context visibility?"
>
> We thank the reviewer for the thoughtful question. MARS’s scheduling benefits are not specific to the choice of OPT‑125M; any lightweight encoder with adequate context handling suffices. In the rebuttal, we have included a BERT-based variant as an alternative predictor to support this point. We also made the analysis more general in the paper by injecting synthetic prediction errors and presenting the results in Figure 7, which show that MARS remains robust even under noisy predictions.
> We selected OPT‑125M for its favorable trade-off between accuracy and inference overhead, and because it supports long contexts (up to 2048 tokens), which is useful for workloads involving multi-turn conversations and interleaved API calls.
> To evaluate robustness across model choices, we trained a BERT‑base variant using the same two-layer MLP head and training setup as OPT‑125M. We include below its accuracy and overhead compared to the OPT‑125M model. We also evaluated MARS using both predictors on the Single-API ToolBench dataset with GPT-J 6B and Vicuna 13B LLMs. The results confirm that MARS’s performance gains are robust across different predictor architectures and LLM backends.
>
> *Prediction Model Accuracy*
>
> | Predictor | Avg. Eval. Loss | Accuracy (%) |
> | --------- | --------------- | ------------ |
> | BERT-base | 0.6492          | 86.46        |
> | OPT-125M  | 0.5614          | 88.06        |
>
> Both models achieve strong accuracy, with OPT‑125M performing slightly better. To further generalize, we injected synthetic noise into the predictions (Figure 7), and MARS continued to outperform FCFS under a wide range of errors.
>
> *Single-API ToolBench (GPT-J 6B Model)*
>
> | Metric          | Stat  | Model     | 3.0 req/s | 4.0 req/s | 5.0 req/s | 6.0 req/s |
> | --------------- | ----- | --------- | --------- | --------- | --------- | --------- |
> | E2E Latency (s) | Mean  | OPT-125M  | 3.15      | 3.31      | 3.73      | 4.22      |
> |                 |       | BERT      | 3.40      | 3.62      | 3.90      | 4.49      |
> |                 | Median| OPT-125M  | 1.77      | 1.97      | 2.23      | 2.71      |
> |                 |       | BERT      | 1.83      | 2.04      | 2.32      | 2.90      |
> |                 | P99   | OPT-125M  | 27.68     | 25.51     | 28.35     | 28.78     |
> |                 |       | BERT      | 24.97     | 25.18     | 25.84     | 27.10     |
> | TTFT (s)        | Mean  | OPT-125M  | 0.06      | 0.10      | 0.14      | 0.24      |
> |                 |       | BERT      | 0.06      | 0.10      | 0.15      | 0.26      |
> |                 | Median| OPT-125M  | 0.03      | 0.03      | 0.04      | 0.07      |
> |                 |       | BERT      | 0.03      | 0.03      | 0.04      | 0.08      |
> |                 | P99   | OPT-125M  | 0.48      | 0.81      | 1.11      | 1.64      |
> |                 |       | BERT      | 0.55      | 0.82      | 1.22      | 1.70      |
> | Throughput      | -     | OPT-125M  | 2.96      | 4.03      | 5.02      | 6.03      |
> | (req/s)         |       | BERT      | 2.96      | 4.03      | 5.02      | 6.03      |
>
> *Single-API ToolBench (Vicuna 13B Model)*
>
> | Metric          | Stat   | Model     | 3.0 req/s | 4.0 req/s | 5.0 req/s | 6.0 req/s |
> | --------------- | ------ | --------- | --------- | --------- | --------- | --------- |
> | E2E Latency (s) | Mean   | OPT-125M  | 3.60      | 3.63      | 3.68      | 3.79      |
> |                 |        | BERT      | 3.73      | 3.73      | 3.96      | 4.09      |
> |                 | Median | OPT-125M  | 2.02      | 2.09      | 2.15      | 2.28      |
> |                 |        | BERT      | 2.01      | 2.15      | 2.22      | 2.36      |
> |                 | P99    | OPT-125M  | 28.40     | 28.26     | 28.41     | 28.44     |
> |                 |        | BERT      | 24.94     | 25.51     | 28.12     | 25.38     |
> | TTFT (s)        | Mean   | OPT-125M  | 0.05      | 0.06      | 0.06      | 0.07      |
> |                 |        | BERT      | 0.05      | 0.06      | 0.06      | 0.08      |
> |                 | Median | OPT-125M  | 0.04      | 0.05      | 0.05      | 0.05      |
> |                 |        | BERT      | 0.04      | 0.05      | 0.05      | 0.05      |
> |                 | P99    | OPT-125M  | 0.16      | 0.22      | 0.31      | 0.33      |
> |                 |        | BERT      | 0.20      | 0.24      | 0.29      | 0.38      |
> | Throughput      | -      | OPT-125M  | 2.96      | 4.03      | 5.02      | 6.03      |
> | (req/s)         |        | BERT      | 2.96      | 4.03      | 5.02      | 6.03      |
>
>
> *Prediction Overhead (Single-API Toolbench)*
>
> OPT‑125M: 10.22 ms/request (updated from 13.7 ms in the paper, which was measured on the multi-API dataset)
> BERT‑base: 7.63 ms/request
>
>
> 2. "Can the authors provide detailed training specifications for the predictor, including dataset size, training epochs, loss convergence, and strategies for handling output length distribution imbalance?"
>
> We thank the reviewer for the request for more details. Below, we summarize how we train the OPT‑125M predictor to classify into 50 output-length bins, how we estimate API response sizes, and how we extend our approach to a BERT-based predictor.
> For each dataset used in our experiments (as listed in the main paper), we split 80% of the data for training and 20% for testing. Each example’s true completion length is discretized into one of 50 equal-width bins (10 tokens each), covering outputs up to 500 tokens.
>
> We use OPT‑125M (125M parameters) as the base model, with all base weights frozen. A LoRA adapter is applied to the classification head, configured with rank = 32, $\alpha = 64$, and a dropout rate of 0.1. Inputs are tokenized up to 2048 tokens (left-padded, following OPT conventions). The final token’s hidden representation is passed through a two-layer MLP (dimensions 512 → 512 → 50) with ReLU and softmax activation.
>
> Training proceeds for 15 epochs using AdamW (learning rate 3e‑4, batch size 64), with a linear warmup over 6% of the total steps. We use cross-entropy loss, log progress to TensorBoard, and save LoRA adapter checkpoints every 5 epochs. The trained predictor achieves an average evaluation loss of 0.561 and classification accuracy of 88.1% on the ToolBench dataset.
> At inference time, our Predictor class loads the trained adapter (e.g., epoch 15), tokenizes the prompt, and predicts the output bin. The output is interpreted as the midpoint of the predicted bin: (predicted_bin×10)+5. The implementation is available in the prediction/ folder of the released codebase, and we will include a detailed description in Appendix C of the camera-ready version.
> Regarding distribution imbalance, we found that binning into equal-width intervals over a 0–500 token range sufficiently spreads examples across bins. We did not apply oversampling or reweighting. In the camera-ready version, we will explore those, particularly for skewed datasets.
>
> 3. "How does the 13.7ms prediction overhead scale with increasing request rates, and what is the break-even point where prediction benefits outweigh the computational cost?"
>
> On the ToolBench with multi-API workload, our OPT‑125M predictor takes 13.7 ms per request on average (Appendix C.6), on ToolBench single API it takes 10.22 ms/request . To reduce this overhead, we can offload prediction to the CPU, without blocking GPU inference, by initiating prediction in parallel as soon as the prompt arrives. We also explore lighter alternatives such as BERT, which we added in the rebuttal, which takes  7.63 ms/request on ToolBench single API dataset. As for the break-even point in end-to-end latency, the 13.7 ms prediction cost is offset by MARS’s scheduling gains, which routinely save tens of milliseconds per request (see Figures 4–5). As long as the latency reduction exceeds 13.7 ms, which holds across all tested arrival rates, the system achieves a net benefit. We will add CPU scaling measurements and a brief discussion of this trade-off in Appendix C.

---

> > ### Comment · Reviewer_H8aQ · 2025-08-01
> >
> > Thank you for your replies to my questions! I have no further questions and keep my score as is.

---

### Official Review · Reviewer_tDDi · 2025-06-30

**Clarity:** 3
**Significance:** 3
**Originality:** 3
**Rating:** 4
**Confidence:** 3

**Summary:**

This paper introduces MARS, a scheduling framework tailored for API-augmented LLMs—language models that interleave generation with external tool or API calls. The key insight is that these requests break the assumptions of traditional scheduling approaches (e.g., token-length-based methods), due to their unpredictable latency and memory behavior. MARS addresses this by (1) predicting each request’s pre-API output length and API call duration, (2) dynamically selecting a memory handling strategy (Preserve, Discard, or Swap) to manage GPU memory usage, and (3) scheduling requests based on their estimated total memory footprint rather than output length. To ensure fairness, it also integrates a starvation prevention mechanism. The system is implemented atop vLLM, supports multi-GPU inference, and is evaluated on ToolBench and INFERCEPT across multiple LLMs (e.g., GPT-J, Vicuna, LLaMA-2 70B). Results show consistent improvements in latency (up to 4.2× reduction in TTFT and 85% reduction in end-to-end latency) over strong baselines like INFERCEPT and vLLM.

**Questions:**

1. Could the authors discuss if MARS's performance is sensitive to the choice of the prediction model? Have preliminary investigations been conducted using other lightweight models for predictions? Providing a brief discussion on this (even if no extensive results are available) would be valuable.

2. Starvation Prevention Threshold Rationale: Could the authors provide a more detailed explanation or sensitivity analysis for the choice of this specific threshold (100)? For example, discussing the trade-offs observed with different thresholds (e.g., how a much lower or higher threshold impacts both tail latency and overall throughput/fairness) would strengthen the justification for this parameter.

3. What should the MARS system do if an API is added or removed?

**Ethical Concerns:**

["NO or VERY MINOR ethics concerns only"]

**Final Justification:**

No further question about author's paper.

**Limitations:**

1. Multi-API Cumulative Memory: The paper explicitly states that the current multi-API approach "does not account for cumulative memory usage," and that "predicting the total number of API calls and their resource usage is challenging, and is left for future work."

2. They acknowledge that "Improving prediction accuracy remains a promising future work," implying that while their simple, lightweight predictor yields benefits, it's not the ultimate solution for prediction.

**Paper Formatting Concerns:**

No major formatting issues.

**Quality:**

3

**Strengths And Weaknesses:**

Strengths:

1. The paper targets a timely and impactful problem—how to efficiently schedule LLM inference with external API calls, a common pattern in modern LLM-powered applications (e.g., tool use, function calling, plugin ecosystems). This makes the work immediately useful for system designers and platform engineers.

2. Extensive evaluations are performed on multiple real-world benchmarks (ToolBench, INFERCEPT) and across multiple LLM backends (GPT-J, Vicuna-13B, LLaMA-2-70B), demonstrating up to 4.2× reduction in TTFT and 85% latency improvement. The ablations are meaningful and help isolate component contributions.

3. The proposed MARS system is implemented atop vLLM, supports multi-GPU inference, and includes practical engineering components such as starvation prevention and per-request memory estimation. The framework is modular and extensible.

Weakness:

1. OPT-125M is mentioned for predictions , more specifics on how it's trained to classify into "50 bins" and how the "average size from the training set for that API class" is used for API response size estimation could be clearer. This could be elaborated in the appendix

2. The experiments are not comprehensive enough, as the study didn't test models other than OPT-125M for predictions.

---

> ### Author Rebuttal · Authors · 2025-07-29
>
> Thank you for recognizing the motivation behind our work and its potential impact on practical applications, especially for system designers and platform engineers.
>  We appreciate your request for additional details on our length predictor. Below, we summarize how we train the OPT‑125M model and estimate per‑API response size. Additionally, as part of the rebuttal, we introduced a BERT-based variant to demonstrate that MARS’s gains are not tied to a specific model architecture.
>
> **OPT‑125M Training Summary**
>
> We selected OPT‑125M for its tradeoff between accuracy and overhead, and because it supports long contexts (2048 tokens), which is userful for workloads that include multiple dialogue turns and API interactions.
>
> *Dataset & Labels*.
> - For each dataset, we split 80 % for training and 20 % for testing.
> - Each example’s true completion length is binned into one of 50 equal-width bins (10 tokens each), covering output lengths in the range 0–500 tokens.
>
> *Model & LoRA Fine-Tuning.*
> - Base model: facebook/opt-125m (125 M parameters). All base weights are frozen.
> - Adapter config: LoRA is applied to the classification head with r = 32, $\alpha = 64$, and a dropout rate of 0.1.
> - Architecture: After tokenizing each prompt (up to 2048 tokens, with left-padding), we extract the final token’s hidden vector and
> pass it through a two-layer MLP (512 → 512 → 50) with ReLU and softmax.
>
> *Training*: 15 epochs using AdamW (learning rate = 3e‑4, batch size = 64) and a linear warmup over 6 % of total steps. We use cross-entropy loss and log training progress in TensorBoard. Adapters are saved every 5 epochs.
>
> *Prediction API*: At inference, our Predictor class loads the trained LoRA adapter (e.g., epoch 15), tokenizes the input prompt, and computes the predicted bin. The output prediction is interpreted as the midpoint of the predicted bin, computed as
> (predicted_bin×10)+5.
>
> *API Response Size Estimation*
> We group examples in the training set by their API name (e.g., Math, QA, Image).
> For each API class, we compute the mean and standard deviation of both response length (in tokens) and API duration.
> At runtime, we use the class mean as the predicted API response size and duration. These values are summarized in Table 1, and we will include a detailed table in Appendix C.6 listing per-class statistics.
>
> The code for this predictor is available in the prediction folder at the provided code repository link and we will add additional description about it to Appendix C based on the reviewer’s comment.
>
> **BERT‑Base Variant Results**
>
> We replace OPT‑125M with a standard BERT‑base encoder, keeping the same MLP head and training setup. We evaluate the BERT-based predictor and compare it against OPT‑125M in terms of accuracy and inference overhead. We also evaluated MARS using both predictors on the Single-API ToolBench dataset with GPT-J 6B and Vicuna 13B LLMs. The results confirm that MARS’s performance gains are robust across different predictor architectures and LLM backends.
>
> *Prediction Model Accuracy*
>
> | Predictor | Avg. Eval. Loss | Accuracy (%) |
> | --------- | --------------- | ------------ |
> | BERT-base | 0.6492          | 86.46        |
> | OPT-125M  | 0.5614          | 88.06        |
>
> Both models achieve strong accuracy, with OPT‑125M performing slightly better. To further generalize, we injected synthetic noise into the predictions (Figure 7), and MARS continued to outperform FCFS under a wide range of errors.
>
> *Single-API ToolBench (GPT-J 6B Model)*
>
> | Metric          | Stat  | Model     | 3.0 req/s | 4.0 req/s | 5.0 req/s | 6.0 req/s |
> | --------------- | ----- | --------- | --------- | --------- | --------- | --------- |
> | E2E Latency (s) | Mean  | OPT-125M  | 3.15      | 3.31      | 3.73      | 4.22      |
> |                 |       | BERT      | 3.40      | 3.62      | 3.90      | 4.49      |
> |                 | Median| OPT-125M  | 1.77      | 1.97      | 2.23      | 2.71      |
> |                 |       | BERT      | 1.83      | 2.04      | 2.32      | 2.90      |
> |                 | P99   | OPT-125M  | 27.68     | 25.51     | 28.35     | 28.78     |
> |                 |       | BERT      | 24.97     | 25.18     | 25.84     | 27.10     |
> | TTFT (s)        | Mean  | OPT-125M  | 0.06      | 0.10      | 0.14      | 0.24      |
> |                 |       | BERT      | 0.06      | 0.10      | 0.15      | 0.26      |
> |                 | Median| OPT-125M  | 0.03      | 0.03      | 0.04      | 0.07      |
> |                 |       | BERT      | 0.03      | 0.03      | 0.04      | 0.08      |
> |                 | P99   | OPT-125M  | 0.48      | 0.81      | 1.11      | 1.64      |
> |                 |       | BERT      | 0.55      | 0.82      | 1.22      | 1.70      |
> | Throughput      | -     | OPT-125M  | 2.96      | 4.03      | 5.02      | 6.03      |
> | (req/s)         |       | BERT      | 2.96      | 4.03      | 5.02      | 6.03      |
>
> *Single-API ToolBench (Vicuna 13B Model)*
>
> | Metric          | Stat   | Model     | 3.0 req/s | 4.0 req/s | 5.0 req/s | 6.0 req/s |
> | --------------- | ------ | --------- | --------- | --------- | --------- | --------- |
> | E2E Latency (s) | Mean   | OPT-125M  | 3.60      | 3.63      | 3.68      | 3.79      |
> |                 |        | BERT      | 3.73      | 3.73      | 3.96      | 4.09      |
> |                 | Median | OPT-125M  | 2.02      | 2.09      | 2.15      | 2.28      |
> |                 |        | BERT      | 2.01      | 2.15      | 2.22      | 2.36      |
> |                 | P99    | OPT-125M  | 28.40     | 28.26     | 28.41     | 28.44     |
> |                 |        | BERT      | 24.94     | 25.51     | 28.12     | 25.38     |
> | TTFT (s)        | Mean   | OPT-125M  | 0.05      | 0.06      | 0.06      | 0.07      |
> |                 |        | BERT      | 0.05      | 0.06      | 0.06      | 0.08      |
> |                 | Median | OPT-125M  | 0.04      | 0.05      | 0.05      | 0.05      |
> |                 |        | BERT      | 0.04      | 0.05      | 0.05      | 0.05      |
> |                 | P99    | OPT-125M  | 0.16      | 0.22      | 0.31      | 0.33      |
> |                 |        | BERT      | 0.20      | 0.24      | 0.29      | 0.38      |
> | Throughput      | -      | OPT-125M  | 2.96      | 4.03      | 5.02      | 6.03      |
> | (req/s)         |        | BERT      | 2.96      | 4.03      | 5.02      | 6.03      |
>
>
> *Prediction Overhead (Single-API Toolbench)*
>
> OPT‑125M: 10.22 ms/request (updated from 13.7 ms in the paper, which was measured on the multi-API dataset)
> BERT‑base: 7.63 ms/request
>
> **Questions:**
> 1. "Could the authors discuss if MARS's performance is sensitive to the choice of the prediction model?"
>
> The close performance results indicate that MARS’s scheduling benefits are not tied to OPT‑125M; any lightweight encoder with sufficient context understanding is effective. We have aimed to make our experimental analysis more general by injecting errors into the predictor (to mimic cases with larger and different errors) and have shown the results in Figure 7.
>
> We will include these additional training details, prediction accuracy results, and the extended evaluation in Appendix C.6 of the camera-ready version.
>
> 2. "Could the authors provide a more detailed explanation or sensitivity analysis for the choice of this specific threshold (100)?"
>
> We appreciate the reviewer’s request for a more detailed explanation of the starvation threshold.
> We included a sensitivity study in Figure 16 (which appears in Appendix C.5 due to lack of space) to justify our choice of a threshold value of 100. This figure compares throughput and P99 latency under several threshold settings.
>
> - No prevention (“off”): Long‐running requests never get preempted, leading to head-of-line blocking.
> - Very low threshold (10): Starvation prevention triggers too aggressively, flattening priority differences. As a result, the benefits of memory-aware scheduling are lost.
> - Very high threshold (1 000 or 10 000): Starvation prevention rarely activates, leading to high tail latency and reduced throughput under load.
> - Threshold = 100: Achieves the best trade-off: starvation prevention triggers often enough to improve tail latency but infrequently enough to retain the efficiency of memory-aware ranking.
>
> We will expand the discussion in Appendix C.5 to make these trade-offs explicit and guide the reader in the main paper to the corresponding curves in Figure 16.
>
>
> 3. "What should the MARS system do if an API is added or removed?"
>
> When an API is added to or removed from a pending request, its predicted handling strategy and rank score must be updated to reflect the new memory and API profile. As soon as the API set changes, we recompute the memory waste (Preserve/Discard/Swap) and update the rank score for that request.
> If the request has not yet begun execution, we remove it from its current position in the waiting queue, update r.score, and re-sort the queue so that its new priority takes effect immediately.
> If the request is already running (i.e., partway through decoding or in an API call), we let it finish under its originally selected strategy. Preempting a request mid-flight would incur additional memory or swap overhead and is unlikely to yield performance gains, especially since API set changes during execution are rare in practice. In the camera-ready version, we will include an experiment to evaluate MARS's performance when API calls are added or removed dynamically.

---

> > ### Comment · Reviewer_tDDi · 2025-08-02
> >
> > Thank you for your replies to my questions! I have no further questions and keep my score as is.

---

### Official Review · Reviewer_VwsY · 2025-07-03

**Clarity:** 2
**Significance:** 2
**Originality:** 3
**Rating:** 4
**Confidence:** 4

**Summary:**

The paper introduces MARS, an inference framework designed to optimize completion latency for API-augmented large language model requests by introducing a predictive, memory-aware scheduling approach.

MARS first predicts each request’s pre-API output size and API call duration to select one of three handling strategies (Preserve, Discard & Recompute, Swap) that minimizes memory waste. It then ranks requests by their predicted cumulative memory footprint and schedules them accordingly.

Implemented on top of vLLM, MARS demonstrate 27–85% end-to-end latency reductions and 4–96% TTFT improvements compared to the existing system, INFERCEPT.

**Questions:**

See the Strengths & Weaknesses section.

**Ethical Concerns:**

["NO or VERY MINOR ethics concerns only"]

**Final Justification:**

The concerns I raised—particularly around the technical descriptions and overhead analysis—are suitably addressed. I believe this paper has only some minor writing issues that can be improved prior to camera-ready submission.

**Limitations:**

yes

**Paper Formatting Concerns:**

N/A.

**Quality:**

2

**Strengths And Weaknesses:**

## Strengths

- The proposed approach is technically sound. It is reasonable to estimate each request’s cost with the API call duration considered. It is intuitive that predicting the cost in advance will be helpful for request scheduling.

- Extensive evaluation across diverse models, datasets, and request rates is conducted, showing substantial performance gains.

- Open-source implementation on vLLM ensures reproducibility and real-world applicability.


## Weaknesses

- Missing critical technical descriptions. While the big picture of the two-step pipeline (e.g., L138-L141) is clear, how each step is implemented remains obscure, making the methodology part difficult to follow

    - How are the output size and API duration? How is the handling strategy predicted, and when does it exactly happen? It is advisable to illustrate these designs with formalized descriptions and diagrams.

    - How are requests ranked, or more specifically, how is the rank index calculated? Formalized descriptions or an algorithm may make this step clearer.

- Lack of overhead analysis. Prediction and scheduling overheads and their impact on throughput are mentioned but not analyzed and deeply quantified.

    - In L312-313, critical issues are mentioned with a phrase and then abruptly point to the Appendix without any analysis, making this paper look incomplete and submitted in a hurry.

---

> ### Author Rebuttal · Authors · 2025-07-29
>
> We thank the reviewer for the thoughtful suggestions and for recognizing the potential impact of our approach on practical applications.
>
> 1. While we acknowledge the concern regarding the clarity of presentation, we emphasize that this is due to space limitations.  In particular, limited space led us to place the pseudocode in Appendix B, which may have made the methodology harder to follow. Below, we clarify the implementation details of each step.
>
> **Step 1: Handling Strategy Assignment**
>
> Upon request arrival, before scheduling or batching, we predict two quantities:
> Pre‑API output length $C$: embed the entire prompt with OPT‑125M (and optionally a BERT‐based variant that we have added as part of the rebuttal), extract the final‐token embedding, and feed it into a two‐layer MLP (512 → 50 bins), trained via cross‐entropy.
> API duration $T_{API}$: parse the prompt for API type and look up its mean and variance of call duration and response size from Table 1.
>
> Using these predictions, we compute the projected memory waste for each strategy via equations (Appendix A.3):
> $$Waste_{Preserve} = T_{API} \cdot C \cdot M$$
> $$Waste_{Discard} = T_{fwd}(C) \cdot (C + C_{other}) \cdot M$$
> $$Waste_{Swap} = 2 \cdot T_{swap}(C) \cdot C_{batch} \cdot M$$
>
>
> We then assign:
> $r.\text{handling} = \arg\min_{s \in \{\text{Preserve}, \text{Discard}, \text{Swap}\}} \text{Waste}_s$
> This logic appears in Appendix B (Lines 149–182 and Algorithm 2–5).
>
> **Step 2: Scheduling via Ranking**
>
> After labeling each request with its handling strategy, we compute its rank as the area under its memory‐consumption curve:
>
> $$\text{rank}(r) = \int_0^{t_{\text{end}}(r)} \text{mem}_r(t) \, dt$$
>
> Where $t_{\text{end}}(r)$ denotes the estimated time when request $r$ completes, and $\text{mem}_r(t)$ follows the piecewise consumption curve from Fig. 3 under the chosen strategy. We schedule the lowest first. The code r.score ← HandlingRanking(r) in Algorithm 1 (App. B, line 14) references this definition.
>
> *Prediction Timing*:
>
> We perform both output‐size and API‐duration predictions immediately upon request arrival, in the scheduler’s first loop, before any batching or memory allocation.
>
> We aim to enhance the presentation of our work in the following ways:
> - Flowcharts illustrate each pipeline step.
> - Rank function definition in the main text.
> - A concise pseudocode in Section 3.
> - Timeline diagram showing when predictions and ranking occur.
>
> We hope these enhancements will make our methodology more self‐contained and straightforward to follow.
>
> 2 - We acknowledge the importance of minimizing scheduling and prediction overhead.
> Our approach adds two steps: computing a score based on predictions and sorting requests by this score during scheduling.
>
> *Prediction Overhead*
>
> We use the OPT‑125M model (125 M parameters) to predict output length. On the ToolBench dataset with multiple API, this prediction takes on average 13.7 ms per request, as reported in Appendix C.6 (with a single API ToolBench dataset takes 10.22ms while the BERT-based predictor takes 7.63 ms per request). To reduce overhead, we can compute predictions on the CPU in parallel with GPU execution. This avoids stalling the GPU pipeline; the prompt is transferred to the CPU upon arrival, and prediction proceeds while the GPU begins LLM inference.
>
> In terms of FLOPs, the predictor is lightweight. For comparison, the smallest LLM we serve is GPT‑J (6 B parameters), which is over 40× larger than OPT‑125M. Even if prediction ran on the GPU, the overhead would be approximately 2% for GPT‑J, 1% for Vicuna‑13B, and 0.18% for Llama‑70B as an upper bound. We will add these numbers and descriptions to Appendix C.
>
> *Scheduling Overhead*
>
> Each incoming request receives a score via a simple arithmetic computation, followed by insertion into a sorted queue. The sorting step operates over small batch sizes (typically tens of requests) and adds negligible latency compared to the millisecond‑scale cost of model inference.
> As a potential optimization, we could update scores only for newly arrived requests or sort every N requests, trading off a small amount of score freshness for efficiency.
>
> *Starvation Prevention*
>
> To ensure throughput and fairness, we implement a starvation counter that increments for requests stuck in the queue. Once the counter exceeds a threshold, we prioritize the request. This adds minimal overhead (one integer update per iteration).
> As part of the rebuttal, we evaluated throughput for MARS, INFERCEPT, and vLLM using Vicuna‑13B on both the SingleAPI and MultiAPI workloads. The results show that MARS consistently achieves higher throughput across all request rates.
>
> SingleAPI result:
>
> | req/s | MARS | InferCept | vLLM |
> |-------|------|-----------|------|
> | 3.0   | 2.94 | 2.94      | 2.94 |
> | 4.0   | 3.99 | 3.78      | 3.57 |
> | 5.0   | 4.78 | 3.54      | 3.25 |
> | 6.0   | 5.29 | 3.61      | 3.09 |
>
>
> MultiAPI result:
>
> | req/s | MARS | InferCept | vLLM |
> |-------|------|-----------|------|
> | 3.0   | 1.98 | 1.40      | 0.98 |
> | 4.0   | 2.04 | 1.36      | 0.83 |
> | 5.0   | 1.74 | 1.33      | 0.69 |
> | 6.0   | 1.63 | 1.35      | 0.67 |
>
>
>
> These clarifications and throughput evaluation, along with overhead quantification, will be added to the main text in the final version.

---

> > ### Comment · Reviewer_VwsY · 2025-08-06
> > **My concerns are addressed**
> >
> > Thank you for your thoughtful rebuttal. I appreciate your clarifications and believe the concerns I raised—particularly around the technical descriptions and overhead analysis—can be suitably addressed in a revision prior to the camera-ready submission if this paper gets accepted.

---

### Note · Authors · 2025-08-13

We thank the reviewers for their thoughtful and constructive feedback. We’re encouraged that multiple reviewers (tDDi, H8aQ) recognized the timeliness and impact of the problem, and that our approach was described as solid (VwsY) and clever (H8aQ). All reviewers (VwsY, tDDi, H8aQ, asX2) noted the breadth and depth of our evaluation across models, datasets, and request rates. Reviewers VwsY and tDDi also appreciated our open-source implementation on top of vLLM, which ensures reproducibility and real-world applicability.

One recurring concern was our choice of prediction model. While the paper used OPT‑125M, we emphasize that MARS’s performance gains are not tied to a specific predictor. In the rebuttal, we introduced a BERT-based variant and compared its performance; the results are included in our responses to the reviewers. In addition, we evaluated throughput for MARS, INFERCEPT, and vLLM using Vicuna‑13B on both the SingleAPI and MultiAPI workloads.

In the revised paper, we will integrate all of the enhancements from our rebuttal, including:

- A BERT‑based predictor with its full evaluation results and discussion.
- Expanded training specifications for the predictor.
- Flowcharts illustrating each step of the pipeline.
- A formal definition of the rank function in the main text.
- Prediction and scheduling overhead discussion.
- Throughput evaluation.

---

### Decision · Program_Chairs · 2025-09-17

**Decision:**

Accept (poster)

**Comment:**

This paper presents an inference framework for scheduling API-augmented LLM requests with the goal of optimizing completion time. Initial reviews were mixed, but after rebuttal and discussion, all reviewers recommended borderline accept. Reviewers agreed that the problem of scheduling API-augmented LLM requests is both timely and important, and noted the extensive results. Most concerns centered on technical details, which appear to have been satisfactorily addressed in the rebuttal. After considering the paper, reviews, and discussion, and given the significance of the problem, the AC recommends acceptance. The authors are encouraged to incorporate the rebuttal clarifications into the camera-ready version.